# Fin whale song evolution in the North Atlantic

**Miriam Romagosa[1]\*, Sharon Nieukirk[2], Irma Cascão[1], Tiago A Marques[3,4], Robert Dziak[5], Jean-Yves Royer[6], Joanne O'Brien[7], David K Mellinger[2], Andreia Pereira[8], Arantza Ugalde[9], Elena Papale[10], Sofia Aniceto[11], Giuseppa Buscaino[10], Marianne Rasmussen[12], Luis Matias[8], Rui Prieto[1], Mónica A Silva[1]**

[1]Institute of Marine Sciences - OKEANOS & Institute of Marine Research - IMAR, University of the Azores, Horta, Portugal; [2]Cooperative Institute for Marine Ecosystem and Resources Studies, Oregon State University, Corvallis, United States; [3]Centre for Research into Ecological and Environmental Modelling, University of St Andrews, St Andrews, United Kingdom; [4]Centro de Estatística e Aplicações, Departamento de Biologia, Faculdade de Ciências, Universidade de Lisboa, Lisboa, Portugal; [5]NOAA Pacific Marine Environmental Laboratory, Hatfield Marine Science Center, Corvallis, United States; [6]CNRS - UBO - UBS - Ifremer, IUEM - Lab. Geo-Ocean, Plouzane, France; [7]Marine and Freshwater Research Centre (MFRC), Atlantic Technological University, Galway, Ireland; [8]Instituto Dom Luiz (IDL), Universidade de Lisboa, Lisboa, Portugal; [9]Institute of Marine Sciences, ICM-CSIC, Barcelona, Spain; [10]Institute for the Study of Anthropic Impacts and Sustainability in the Marine Environment of the National Research Council of Italy (CNR-IAS), Torretta Granitola, Italy; [11]Akvaplan-niva, Tromsø, Norway; [12]University of Iceland Research Centre in Húsavík, Húsavík, Iceland

**\*For correspondence:**
m.romagosa4@gmail.com

**Competing interest:** The authors declare that no competing interests exist.

**Abstract** Animal songs can change within and between populations as the result of different evolutionary processes. When these processes include cultural transmission, the social learning of information or behaviours from conspecifics, songs can undergo rapid evolutions because cultural novelties can emerge more frequently than genetic mutations. Understanding these song variations over large temporal and spatial scales can provide insights into the patterns, drivers and limits of song evolution that can ultimately inform on the species' capacity to adapt to rapidly changing acoustic environments. Here, we analysed changes in fin whale (*Balaenoptera physalus*) songs recorded over two decades across the central and eastern North Atlantic Ocean. We document a rapid replacement of song INIs (inter-note intervals) over just four singing seasons, that co-occurred with hybrid songs (with both INIs), and a clear geographic gradient in the occurrence of different song INIs during the transition period. We also found gradual changes in INIs and note frequencies over more than a decade with fin whales adopting song changes. These results provide evidence of vocal learning in fin whales and reveal patterns of song evolution that raise questions on the limits of song variation in this species.

## Editor's evaluation

This study is a valuable contribution to our understanding of vocal variation in acoustic displays of male baleen whales, part of a developing story about cultural change in songs in species other than the relatively well studied humpback whales. The authors present solid evidence of changes at various timescales in 20-Hz song note intervals and call center frequency over decadal time scales and large spatial scales.

## Introduction

Animal songs, often used as acoustic sexual displays, can change within and between populations through different evolutionary processes. These processes can be selective (i.e. sexual, cultural, or natural selection), favouring song changes that confer advantages to singers, or non-selective (i.e. cultural or genetic drift), causing random changes in songs. Both selective and non-selective processes may result in rapid and gradual population-wide shifts in the structure, complexity, frequency, and temporal properties of songs (*Garland et al., 2011*; *Otter et al., 2020*; *Whiten, 2019*; *Williams et al., 2013*), although song evolution can also be constrained by the species' genetic variation and mechanical design (*Podos et al., 2004*).

The best-known examples of song evolution are found among songbirds, which field studies started decades ago and have led to extensive literature on the topic. Songs from many songbirds are culturally transmitted through vocal learning, wherein animals learn to sing by hearing and imitating conspecifics (*Williams, 2021*). Vocal learning and specific patterns of dispersal are largely responsible for the geographic variation found in songs of many bird species (*Podos and Warren, 2007*). The formation of local dialects is, in part, a consequence of certain mechanisms of song learning (i.e., copying 'errors') that generate vocal novelties (*Podos and Warren, 2007*). Learned songs may also undergo rapid evolutions within populations, basically because cultural novelties can emerge more frequently than genetic mutations (*Wilkins et al., 2013*). One example of rapid song evolution is found in the white-throated sparrow (*Zonotrichia albicollis*), in which a new doublet ending song spread across the North America continent in less than 20 years, completely replacing the established triplet-ending song. This fast spread is believed to have occurred because birds singing the old and new songs overwintered in the same grounds and learned from each other. Although it remains unclear why the new song overturned the old song, one possible explanation for this rapid song transition is that certain innovations are adopted non-randomly by all males to maintain female interest (*Otter et al., 2020*). Birdsong properties can also show a gradual directional evolution in response to specific evolutionary process. A clear example are birds from urban areas, which song elements increase in frequency (Hz) in response to noisy environmental conditions (*Slabbekoorn, 2013*). Directional song evolution can also cause directional song changes (e.g. faster trill rates, broader frequency bands, and lower frequency trills) driven by sexual selection operating through male-male interactions, mate choice by females or both (*de Kort et al., 2009*; *Illes et al., 2006*; *Williams et al., 2013*). Yet, all these song variations are constrained by the singers' morphological (e.g. beak shape or body size) and neurological limitations that sometimes can hinder the animals' adaptation to rapid human induced changes in the environment (e.g. urban noise; *Luther and Derryberry, 2012*; *Podos et al., 2004*).

A parallelism to birdsong evolution can be found in the marine realm. Songs from humpback whales (*Megaptera novaeangliae*) differ across ocean regions (*Winn et al., 1981*), evolve gradually over time (*Payne et al., 1983*; *Payne and Payne, 1985*) and can go through revolutionary changes (*Noad et al., 2000*). During song revolutions, a population song type is rapidly replaced by a novel song type introduced from a neighbouring population (*Garland et al., 2011*; *Noad et al., 2000*). Most authors agree that these spatial and temporal patterns in humpback whale song changes can only be explained by vocal learning (*Garland et al., 2011*; *Janik and Knörnschild, 2021*; *Noad et al., 2000*; *Tyack, 2008*). Yet, the learning capacity of novel songs in humpback whales may be limited because song complexity always decreases in each revolutionary event (*Allen et al., 2018*).

Compared to humpback whales and most songbirds, fin whales (*Balaenoptera physalus*) produce simpler songs consisting of a stereotyped repetition of a few low-frequency note types. These songs are also believed to act as mating displays (*Thompson et al., 1992*; *Watkins et al., 1987*), because they are produced by males (*Croll et al., 2002*) and intensify during the breeding season (*Lockyer, 1984*; *Širović et al., 2013*; *Thompson et al., 1992*; *Watkins et al., 1987*). In this species, the song inter-note interval (INI) is the most distinctive parameter between regions (*Castellote et al., 2012*; *Delarue et al., 2009*; *Hatch and Clark, 2004*; *Širović et al., 2017*; *Watkins et al., 1987*) and has been used to differentiate stocks and populations (*Castellote et al., 2012*; *Delarue et al., 2009*; *Morano et al., 2012*; *Širović et al., 2017*; *Wood and Širović, 2022*). Previous studies showed that fin whale song INIs differed between western, central, and eastern North Atlantic areas, as well as between these and the Mediterranean Sea (*Castellote et al., 2012*; *Delarue et al., 2009*; *Hatch and Clark, 2004*; *Morano et al., 2012*). These results partially agree with genetic data that indicate significant levels of heterogeneity in the mitochondrial DNA between the Mediterranean Sea, the eastern (Spain),

and the western (Gulf of Maine and Gulf of St Lawrence) North Atlantic; however, samples from West Greenland and Iceland could not be assigned to either of the two North Atlantic areas, suggesting a mixture of subpopulations in these feeding grounds (*Bérubé et al., 1998*). Another large-scale study combining fin whale genetic and song data from the Northeast Pacific, North Atlantic, and Mediterranean Sea showed that acoustic differentiation among fin whales were not always reflected in estimates of genetic divergence (*Hatch and Clark, 2004*). These authors concluded that differences in songs may reflect differences in fin whale movements and/or social and vocal behaviours that occur at shorter timescales than genome evolution. In fact, fin whale song INIs can change abruptly from one year to the next in the same region (*Delarue et al., 2009*; *Hatch and Clark, 2004*; *Helble et al., 2020*; *Morano et al., 2012*; *Širović et al., 2017*) and have been progressively changing over time in different ocean regions (*Best et al., 2022*; *Helble et al., 2020*; *Leroy et al., 2018a*; *Weirathmueller et al., 2017*). Also, the center frequencies of two fin whale song components, the 20 Hz note and the higher frequency (~130 Hz) upsweep (hereafter HF note; *Hatch and Clark, 2004*), have been decreasing gradually over the last decade in different ocean basins (*Leroy et al., 2018a*; *Weirathmueller et al., 2017*; *Wood and Širović, 2022*).

Currently, we do not understand the mechanisms and drivers of fin whale song variations nor how these variations may be affected by the species' physiological and morphological constraints of vocal performance. Broad-scale studies matching the known scales of fin whale natural history and ecology can shed light into the species' population structure and demography, even before genetic differentiation is evident, and elucidate patterns, drivers and limits of song evolution that can ultimately inform on the species' capacity to adapt to human-induced changes in their acoustic habitats (e.g. anthropogenic noise and climate change).

Our study attempts to address these issues by investigating changes over two decades of three fin whale song parameters (INIs and peak frequencies of the 20 Hz and HF note types) in a wide area of the North Atlantic Ocean. Our work provides evidence of social learning in fin whale songs and shows: (i) a rapid evolution in song INIs across a vast area of the central North Atlantic in just four singing seasons, with the existence of hybrid songs (including both INIs) and a clear geographic gradient of song INIs during the transition period; (ii) a gradual evolution of song parameters showing an increase in INIs and a decrease in frequencies of the 20 Hz and HF notes over more than a decade in the central and eastern North Atlantic; and (iii) the adoption of both rapid and gradual song changes by fin whales from a wide region. We conclude by discussing song changes under the scope of cultural transmission, song function and the limits of song variation.

## Results

The processing across all acoustic data resulted in 379 songs, from which 39680 INIs and its corresponding note frequencies were measured, and 143 songs, from which 9185 HF note peak frequencies were measured (*Supplementary file 1a*; 'Materials and methods'). The greatest numbers of INIs came from the SE and Azores locations in the Oceanic Northeast Atlantic (ONA) region, with ~32% and ~1% respectively. Contributions from the remaining locations of the ONA region ranged from 3% to 7%, while contributions from locations outside the ONA region ranged from 0.6% to 7%. Among the locations in which the measurement of the HF note was possible, the Azores and SE Greenland locations contributed the most (~46% and ~22% respectively; *Table 1*).

### Transition in song INIs in the Oceanic Northeast Atlantic region

Results showed a rapid shift in song INIs in the SE location from the ONA region (*Figure 1A*), previously noted by *Nieukirk et al., 2011*, where songs with 19 s INIs were completely replaced by songs with 12 s INIs in just four singing seasons (*Figure 1B and C*). In 1999 and 2000, the 19s-INI song was the only song present in this location. By 2004, the 19s-INI song had disappeared from this location (*Figure 1B*) and was not detected in any of the sampled regions from 2006 to 2020, except from a single song in 2008 Figure 3A. During the transition period, 12s- and 19s INI-songs co-existed and there was a notable percentage of songs containing both INIs, which we refer as 'hybrid songs'. Hybrid songs showed two INIs in variable ratios and no apparent pattern, either mixed within the same song sequences (i.e. series of consecutive 20 Hz notes separated by periods of silence; *Watkins*

**Table 1.** Sampling information and effort.

For each location within each region this table shows: sampled period, duty cycle, sampling rate (Samp. rate), total number of recording hours (Rec. hours), number of measured inter-note intervals (INIs) (Num. INIs), contribution to total number of INIs measured (Contr. INIs), number of measured high frequency (HF) note peak frequencies (Num. HF note) and percent contribution to total number of measured HF note peak frequencies (Contr. HF note).

| Region | Location | Sampled period | Duty cycle (%) | Samp. rate (Hz) | Rec. hours | Num. INIs | Contr. INIs (%) | Num.HF note | Contr. HF note (%) |
|---|---|---|---|---|---|---|---|---|---|
| SE Greenland | SE Greenland | 01/10/2007 - 14/03/2008 | Cont. | 2000 | 4392 | 2841 | 7.1 | 2040 | 22.2 |
| SE Iceland | SE Iceland | 04/01/2007 - 31/03/2007 | Cont. | 4000 | 2088 | 291 | 0.7 | 169 | 1.9 |
| Celtic Sea | North Porcupine | 01/10/2015 - 03/11/2016 | 10 | 2000 | 520.8 | 286 | 0.7 | 215 | 2.3 |
| | South Porcupine | 01/10/2015 - 03/11/2016 | 10 | 2000 | 520.8 | 674 | 1.7 | 627 | 6.9 |
| | NE | 01/10/2002 - 31/03/2003 | Cont. | 250 | 4344 | 2650 | 6.7 | NA | NA |
| | NW | 01/10/2002 - 31/03/2003 | Cont. | 250 | 4344 | 2754 | 7.1 | NA | NA |
| | CE | 01/10/2002 - 31/03/2003 | Cont. | 250 | 4344 | 2817 | 7.1 | NA | NA |
| | CW | 01/10/2002 - 31/03/2003 | Cont. | 250 | 4344 | 2930 | 7.3 | NA | NA |
| | | 01 –31/01/2006; 01-31/01/2007;01-31/01/2008 | Cont. | 2000 | 2232 | 573 | 1.4 | NA | NA |
| | | 01/10/2008 - 06/03/2011 | 10 | 2000 | 650.4 | 749 | 1.8 | 1335 | 14.5 |
| | | 15/10/2011 - 06/03/2012 | 43 | 2000 | 1497.6 | 1017 | 2.5 | 677 | 7.3 |
| | | 01/10/2012 - 18/10/2012 | 29 | 2000 | 122.4 | 12 | 0.0 | 216 | 2.3 |
| | | 23/02/2017 - 31/03/2020 | 25 | 2000 | 2136 | 2382 | 6 | 2036 | 22.1 |
| | Azores | Total | | | 6638.4 | 4733 | 11.9 | 4264 | 46.4 |
| ONA | SE | 08/02/1999 - 31/03/2005 | Cont. | 110 | 31704 | 12996 | 32.7 | NA | NA |
| | SW | 31/12/2002 - 31/03/2003 | Cont. | 110 | 2184 | 1517 | 3.8 | NA | NA |
| | | 01/12/2007 - 29/02/2008 | Cont. | 100 | 2184 | 818 | 2.1 | NA | NA |
| SW Portugal | SW Portugal0 | 01/10/2015 - 31/03/2016 | 20 | 2000 | 878.4 | 1195 | 3.1 | 1294 | 14 |
| Canary Islands | Canary Islands | 01/11/2014 - 29/02/2015 | Cont. | 100 | 2160 | 1991 | 5.1 | NA | NA |
| | Svalbard | 02/10/2014 - 31/01/2016 | Cont. | 500 | 2882 | 260 | 0.6 | 165 | 1.8 |
| Barents Sea | Vesterålen | 01/01/2018 - 28/02/2018 | Cont. | 400 | 1416 | 927 | 2.3 | 411 | 4.5 |
| Total | | | | | 74944.4 | 39680 | 100 | 9185 | 100 |

et al., 1987) or found separated in different sequences from the same song. The singing season with the most hybrid songs was 2002/2003, when there were ~30% of hybrids.

Six locations of the ONA region, with simultaneous data from the 2002/2003 singing season, were used to analyse the spatial pattern in song INIs. In this period, the prevalence of songs with each INI type showed a clear spatial gradient across the entire ONA region. The 19s-INI song largely dominated in the SW ONA, with only 9% of hybrid songs, and no detection of 12s-INI songs. The proportion of 19s-INI songs decreased progressively to the east, reaching 0–4% in the easternmost locations (CE and NE), where the 12s-INI songs were prominent (90 and 83%). Hybrid songs were more abundant (17–23%) at central ONA (NW, CW, and SE) than in easternmost locations (NE and CE; 10–13%; *Figure 2*).

## Gradual changes in song INIs and notes frequencies

After the song transition, we found a gradual change in three fin whale song parameters over more than a decade, with most regions fitting the trend. The only exception was the Barents Sea where INIs differed from the rest of the sampled area showing a bimodal pattern. From 2006 to 2021, INIs

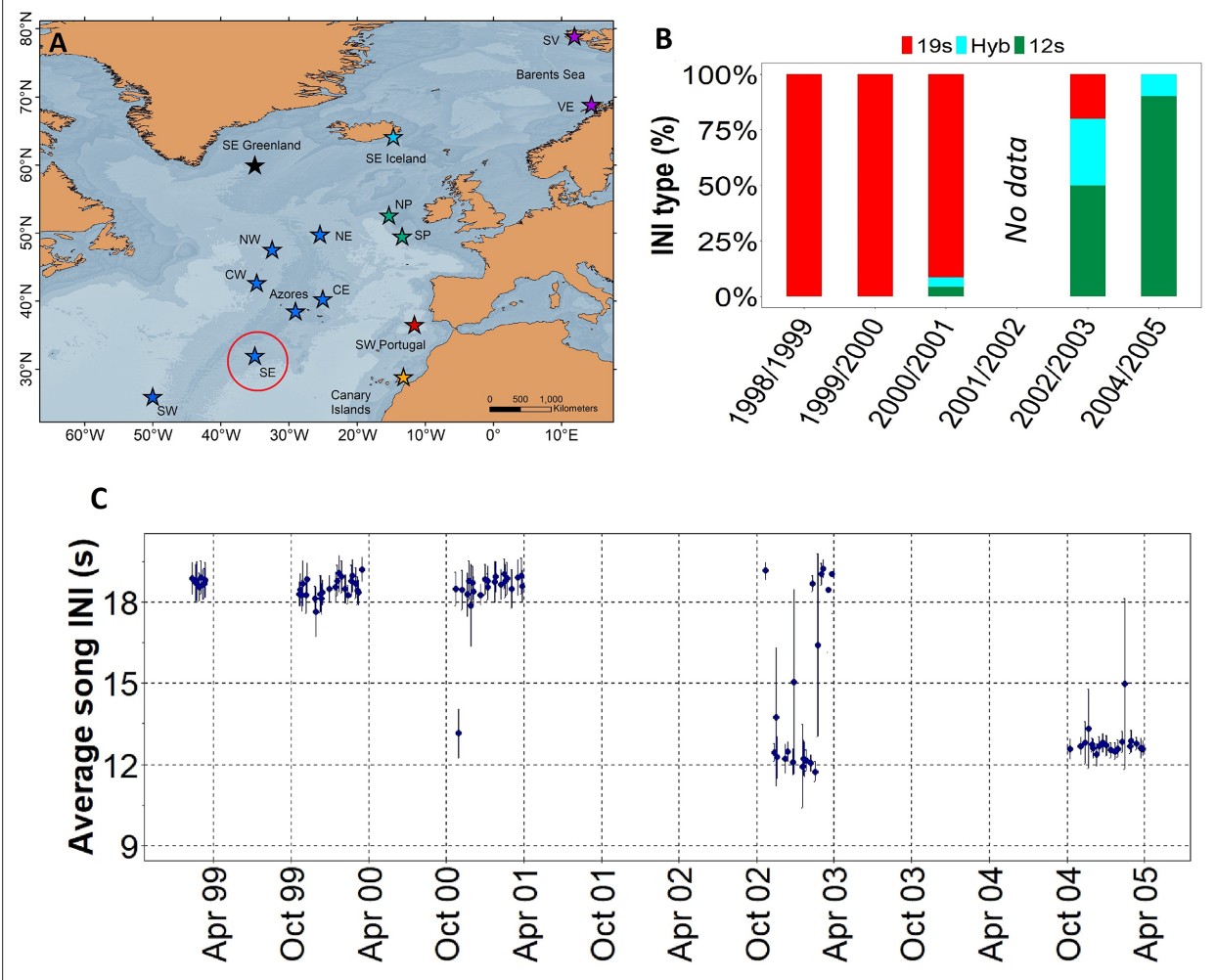

**Figure 1.** Transition in song INIs in the Oceanic Northeast Atlantic region. (**A**) Map showing the SE location (red circle) of the Oceanic Northeast Atlantic region. (**B**) Percentage of songs with each inter-note interval (INI) type (19 s, hybrid (hyb), 12 s) in this location during the song INI shift in 1999 – 2005. (**C**) INIs from 1999 to 2005 for this same location. Points represent mean values per song and error bars are standard deviations.

increased at 0.21 s/yr (Adj. R-sq.=0.4; p<0.001) (*Figure 3A* and *Figure 3—figure supplement 1*). Peak frequencies of the 20 Hz note decreased at a rate of –0.06 Hz/yr (Adj. R-sq.=0.1 from 2009 to 2020; p<0.001) (*Figure 3B* and *Figure 3—figure supplement 1*) while peak frequencies of the HF note decreased at a rate of –0.35 Hz/yr (Adj. R-sq.=0.8; p<0.001) from 2007 to 2020, with all regions fitting the trend including the Barents Sea region (*Figure 3C* and *Figure 3—figure supplement 1*).

## Differences in fin whale song parameters between regions

When comparing data from different regions (SE Iceland, SE Greenland, ONA, SW Portugal, Canary Islands, Barents and Celtic Sea) with simultaneous recordings (i.e. in the same singing season) results showed unimodal overlapping distributions in INIs and HF note peak frequencies (*Figure 4*). The only exception was the Barents Sea region, where INIs differed from the Canary Islands in 2014/2015 (Barents Sea:~9 s and ~14 s; Canary Islands:~15 s), from SW Portugal and the Celtic Sea in 2015/2016 (Barents Sea:~10 s and ~15 s; SW Portugal and Celtic Sea:~15 s) and from the ONA region in 2017/2018 (Barents Sea:~10 s and 16 s; ONA:~16 s; *Figure 4A*).

## Discussion

The rapid and gradual changing patterns of different fin whale song parameters reported here for a wide area of the central and eastern North Atlantic provides evidence of vocal learning in this species.

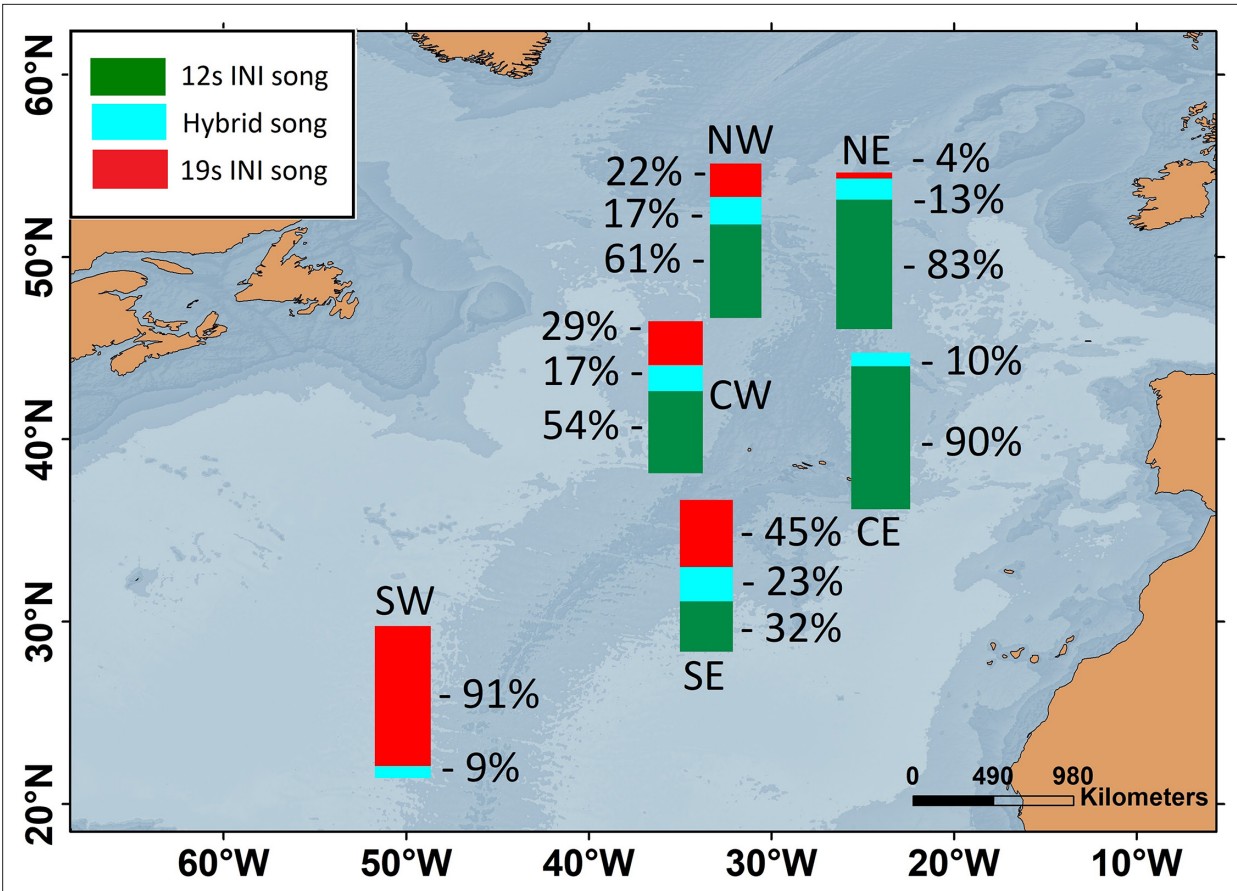

**Figure 2.** Map showing the percentages of fin whale songs with each inter-note interval (INI) type for six locations within the Oceanic Northeast Atlantic region during the 2002/2003 singing season.

Decoupled variations in song INIs and frequency (i.e. INIs changed abruptly but frequencies did not) reveal the complex interplay between different selective pressures and shed some light on the potential limits of song variation.

The rapid replacement of fin whales' song INIs (from 19s to 12s) described here for the ONA region cannot be explained by environmental causation. The shift in INIs found in the ONA region seemed to occur simultaneously at northern feeding grounds, in the so-called Northeast North Atlantic (NENA) region (*Hatch and Clark, 2004*). This variation in INI patterns during the same singing season between neighbouring locations within ONA, together with an identical shift in INIs documented for the same period in the environmentally distant NENA region (*Hatch and Clark, 2004*), strongly suggest that the transition in INIs was not a response to local acoustic environments. Fin whale song INIs are regionally distinct (*Castellote et al., 2012*; *Constaratas et al., 2021*; *Delarue et al., 2009*; *Hatch and Clark, 2004*; *Morano et al., 2012*; *Pereira et al., 2020*; *Širović et al., 2017*; *Víkingsson and Gunnlaugsson, 2006*) and the shift in INIs found in our study could have been caused by a population replacement. However, if this was the case, we would not find hybrid songs containing both INIs during the transition period, as the new song pattern would simply substitute the former, as documented for fin whale songs off Southern California (*Širović et al., 2017*). Multiyear and seasonal alternation of different fin whale song INIs, with presence of hybrid songs, have also been reported in two regions of the Northwest Atlantic. In both cases, authors suggest INI shifts occurred within the same population (*Delarue et al., 2009*; *Morano et al., 2012*). Thus, we suggest that the rapid turnover of fin whale song INIs along a spatial gradient in the ONA region, with males adopting the new song INI, and the existence of hybrid songs, is the result of cultural transmission, the social learning of information or behaviours from conspecifics (*Rendell and Whitehead, 2001*).

Our study also shows that fin whale song INIs from the distant Barents Sea region differ from the rest of the sampled area (central and eastern North Atlantic). Geographic differentiation in song INIs

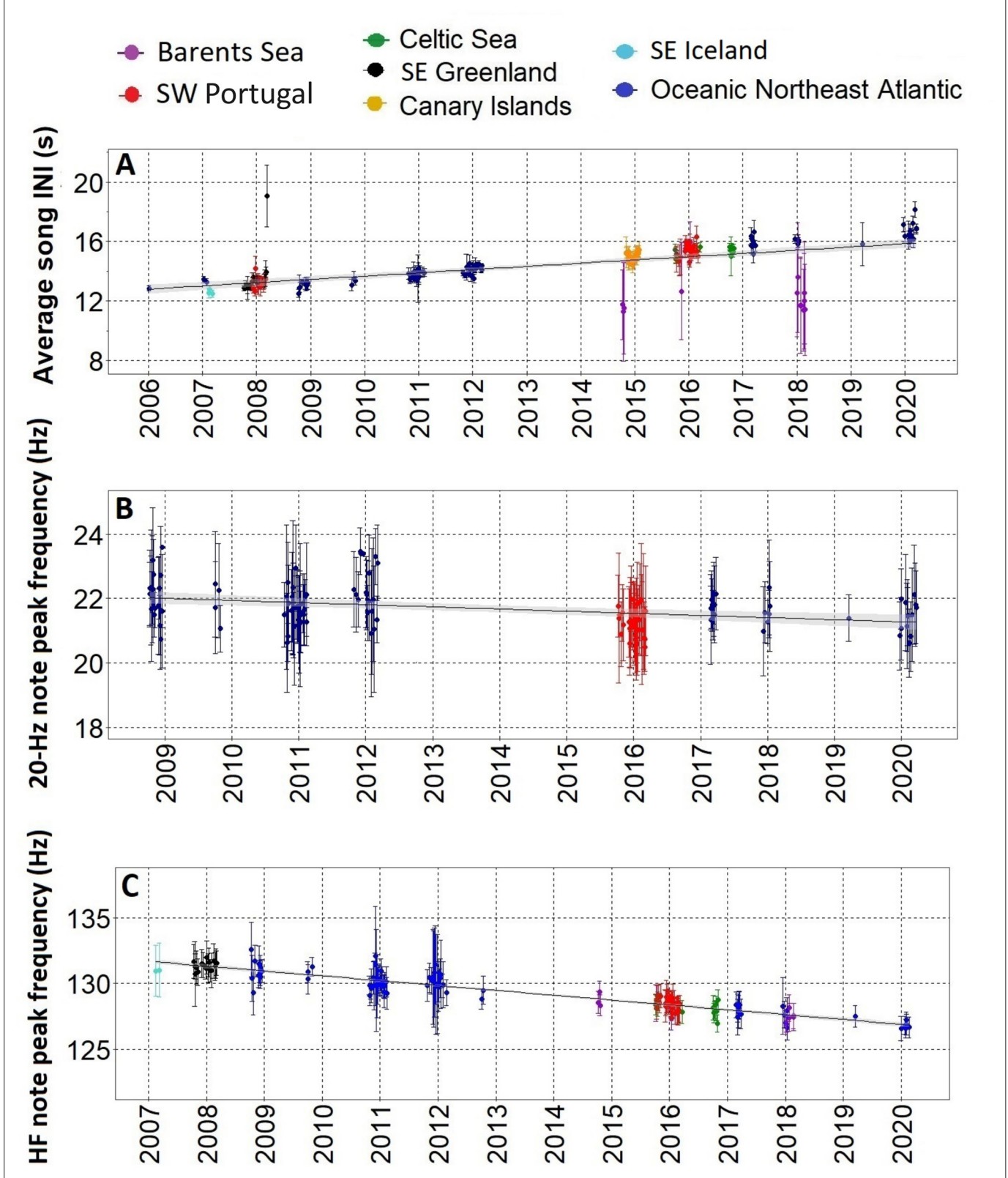

**Figure 3.** Gradual changes in song INIs and notes frequencies. (**A**) Inter-note intervals (INIs) from 2006 to 2020 for all regions sampled. INIs increased at a mean rate of 0.21 s/yr. (**B**) Peak frequencies of the 20 Hz note for SW Portugal (2015/2016) and Azores locations (Oceanic Northeast Atlantic region) sampled with Ecologic Acoustic Recorders (**Lammers et al., 2008**) (2008–2020); these changed at a mean rate of –0.06 Hz/yr. (**C**) Peak frequencies of the

*Figure 3 continued on next page*

*Figure 3 continued*

High Frequency (HF) note for all regions sampled; these changed at a mean rate of –0.36 Hz/yr. Points represent average values per song, error bars are standard deviations and black lines represent the fitted linear regression model with confidence intervals in shadowed grey.

The online version of this article includes the following figure supplement(s) for figure 3:

**Figure supplement 1.** Linear model validation plots for fin whale song inter-note intervals (INIs) (**A**), 20 Hz note peak frequencies (**B**) and HF note peak frequencies (**C**).

**Figure supplement 2.** Peak frequencies of the 20 Hz note for the regions sampled and equipment used: ARs (circle), and OBS (triangle).

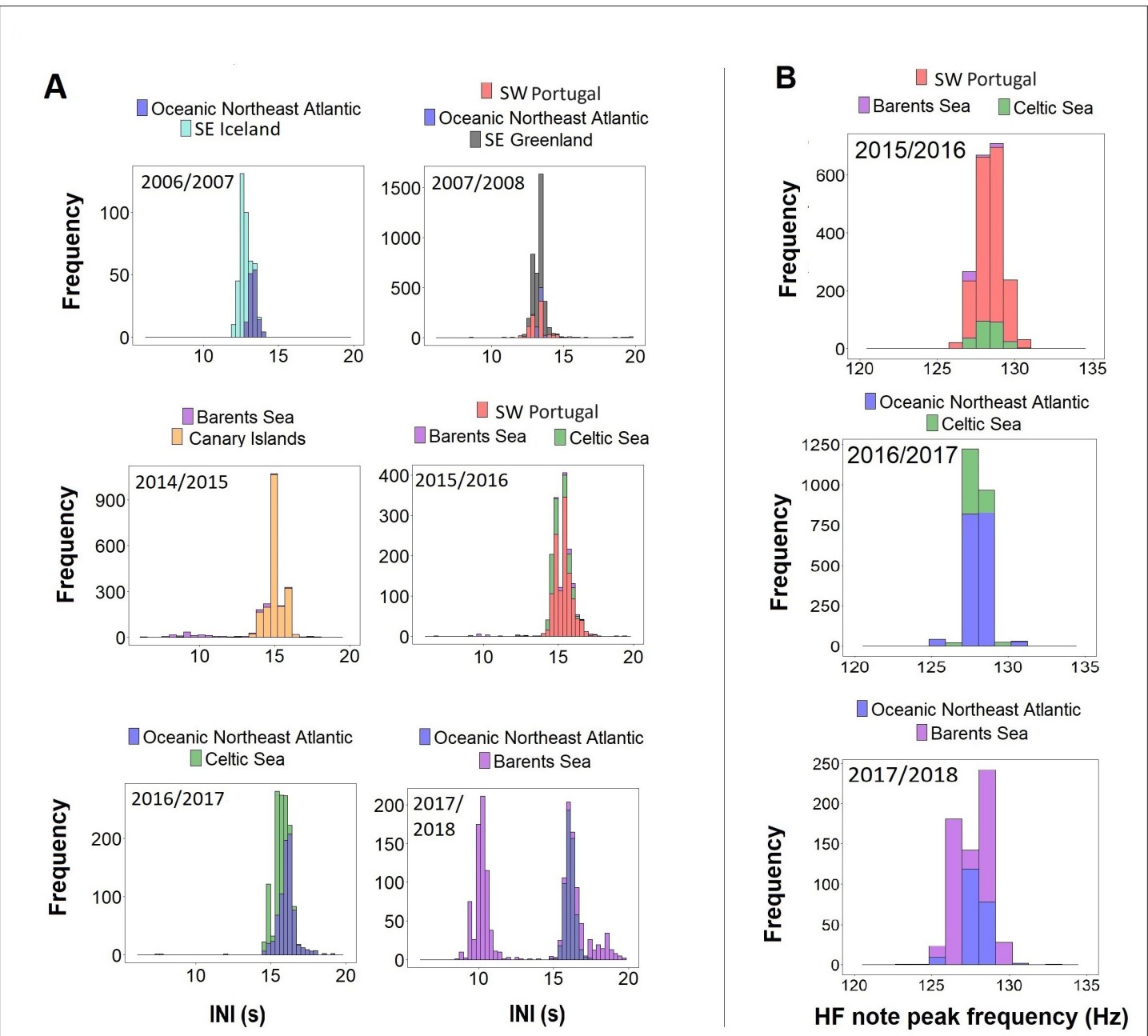

**Figure 4.** Histograms of (**A**) inter-note intervals (INIs) and (**B**) higher frequency (HF) note peak frequencies by singing season (Oct-Mar) from regions with concurrent data.

(*Hatch and Clark, 2004*) with fin whales within a certain area conforming the same INI has been widely documented (*Castellote et al., 2012*; *Delarue et al., 2009*; *Hatch and Clark, 2004*; *Širović et al., 2017*; *Wood and Širović, 2022*). Bird songs also vary geographically, and this variation can be largely attributed to their ability of learning to vocalize through imitation (*Kroodsma, 2004*; *Podos and Warren, 2007*). When songbirds learn their songs from models (i.e. conspecifics) inhabiting the same geographic area where they set their breeding territories, local similarities in song structure can arise (i.e. dialects). This learning can occur after dispersal with birds learning or retaining the dialects sang in the breeding grounds where they set (*Nelson et al., 2001*). Thus, in most species in which vocal learning occurs, the distribution of learned songs may reflect the social interactions among birds, not the genetic structure of the populations (*Kroodsma, 2004*). A decoupling between patterns of cultural (songs) and genetic variation has also been reported for fin whales (*Hatch and Clark, 2004*), further suggesting that song INIs may be socially learned in this species. Learning of novel rhythms (i.e. INIs; *Vernes et al., 2021*) can also be found in sperm whales (*Physeter macrocephalus*), which can match their clicks to the rhythm of a ship echosounder (*Backus and Schevill, 1996*), and use codas (i.e. rhythmic patterns of clicks) that are unique to each vocal clan and are socially learned (*Rendell and Whitehead, 2003*). Fin whales may also be able to learn songs from other populations that not only differ in their INIs but also in their note composition (*Helble et al., 2020*).

After the song transition from 1999 to 2005, we found a gradual increase in song INIs along with a decrease in peak frequencies of the 20 Hz and HF notes. These findings are in line with the gradual trends of decreasing frequencies (*Best et al., 2022*; *Leroy et al., 2018b*; *Weirathmueller et al., 2017*) and increasing INIs (*Best et al., 2022*; *Morano et al., 2012*; *Širović et al., 2017*; *Weirathmueller et al., 2017*) described for fin whale songs in other ocean basins and in the Mediterranean Sea. Contrarily to the rapid changes in INIs, a global-scale process of cultural transmission cannot explain these directional changes. First, changes in INIs and frequencies occur at different rates in different oceans and there is no convergence in song acoustic characteristics across populations (*Leroy et al., 2018a*; *Širović et al., 2017*; *Weirathmueller et al., 2017*). Second, a similar pattern of decreasing frequencies and increasing INIs has been described for blue whale (*B. musculus*) songs (*Jolliffe et al., 2019*; *Malige et al., 2020*; *McDonald et al., 2009*), and decreasing frequencies have been reported for bowhead whales (*Balaena mysticetus*) calls (*Thode et al., 2017*). Such gradual song changes in multiple species and different ocean basins suggest an adaptation to a common selective pressure, which does not mean that within-region conformity in song characteristics does not result from cultural transmission. Mathematical modelling of the linear decrease in blue whale song frequencies suggest a simultaneous effect from two selection processes: conformity and sexual selection (*Malige et al., 2022*). Conformity would occur because individuals would be more likely to share variants of a cultural trait with nearby individuals than with more distant ones. This could be caused either by a conformist bias, which occurs when individuals select common variants from those available more often than would be expected by chance, or by more simple processes, such as only learning from nearby individuals (*Morgan and Laland, 2012*). Sexual selection would drive males to sing lower frequency songs than other whales, presumably because females prefer bigger males that are able to sing lower pitch songs (*Malige et al., 2022*). Increased blue whale body size in a post-whaling recovery scenario has also been proposed as a potential explanation for this species' song changes; yet blue whale body size distributions should have returned to near pre-whaling values by now and song frequencies continue to decrease (*McDonald et al., 2009*). Also, it is very unlikely that changes in whale body size evolved in such a straight line at this timescale (*Malige et al., 2022*). Fin whale songs may evolve in a similar way as blue whale songs do, but so far, none of the proposed hypotheses can convincingly explain the slow frequency song changes in these species (*McDonald et al., 2009*; *Thode et al., 2017*). Large-scale and long-term datasets would help understanding if fin whale song INIs and frequencies are constantly evolving or started changing recently in response to a new driver.

The rapid and gradual evolution of fin whale song parameters found in this, and other studies (*Hatch and Clark, 2004*; *Širović et al., 2017*; *Weirathmueller et al., 2017*), resemble the patterns of song evolution of some bird species and humpback whales. Evidence from songbirds suggest that these different trajectories in song evolution (rapid versus gradual) occur within certain boundaries because learned songs are subject to a combination of strong stabilizing selection and underlying genetic variation that prevent incremental change for long periods of time (*McEntee et al., 2021*). In humpback whales, song complexity increases as songs evolve gradually over time, but decreases

when revolutions occur (i.e. periods of rapid song changes), suggesting that learning capacities in this species are limited (*Allen et al., 2018*). After the rapid shift in fin whale song INIs, from 19s to 12s, a gradual reset towards the 19s-INIs seems to be occurring in all sampled areas, except from the Barents Sea. In the northwest Atlantic Ocean, rapid shifts in fin whale song INIs occurred between 15 s and 9 s (*Delarue et al., 2009*; *Morano et al., 2012*). Perhaps, like in birdsongs and humpback whales, changes in fin whale song INIs are also limited by learning constrains and genetic predispositions. Our results show that fin whale song INIs from the Barents Sea region differ from the rest of the sampled area. Yet, satellite tracking data from 2015 to 2019 showed that some fin whales summering in Svalbard (Barents Sea) migrate to the SW Portugal region in fall and winter (*Lydersen et al., 2020*), so some degree of mixing between males from these two acoustic populations occur. Also, a recent study from Svalbard revealed that fin whale song INIs differed between singing seasons, which suggests that either fin whales from that area switch between INIs or different populations use the area sequentially (*Papale et al., 2023*). Investigating the changing patterns of fin whale song INIs in these two regions (Barents seas and SW Portugal) may shed some light on the learning mechanisms of song INIs and the limitations of its variability.

Compared to INIs, fin whale song frequencies of the 20 Hz and HF notes do not vary abruptly but only gradually. Fundamental frequencies of this species' songs seem constrained by the optimisation of long-range communication in pelagic environments (*Clark and Garland, 2022*; *Payne and Webb, 1971*). This song frequency limitation may be an adaptation first, to a dispersed and open water distribution of this species during the breeding season (*Edwards et al., 2015*; *Nieukirk et al., 2004*) and second, to match a particular frequency band with low levels of noise in deep waters (i.e. a quiet window in frequency) (*Clark and Garland, 2022*; *Curtis et al., 1999*). Comparatively, humpback and right whales (*Eubalaena* spp.) aggregate in coastal breeding grounds (*Clapham, 2018*; *Kenney, 2009*) and use higher frequency songs and calls that transmit better in shallow environments (quiet window: 100–400 Hz) (*Clark and Ellison, 2004*) and do not need to reach distant conspecifics (*Clark, 1982*; *Clark and Garland, 2022*). Therefore, the acoustic environment during the mating season and the species' breeding behaviour could constrain variation in song frequencies to keep them within the quiet window. In addition, the animals' physiology can constrain song frequency variation. In birds, the ability to produce low-frequency songs is linked to body size (*Ryan and Brenowitz, 1985*). If fin whale song frequencies continue to decrease, it can potentially reach the physiological limits of sound production. These limits in song variation can compromise song function and ultimately male fitness when the acoustic habitat in which these songs evolved is changing too rapidly to adapt. For example, the vocal adaptation ability of birds in urban environments (e.g. increasing song frequencies) affect the detection by receivers. If birds are not able to avoid the masking of their songs by noise, this may difficult the establishment and defence of a territory that can ultimately affect their fitness (*Habib et al., 2007*; *Luther and Derryberry, 2012*). Similarly, the constraints in fin whale song frequency may limit adaptation to an increasingly noisy environment. Shipping noise, the major source of ocean noise, overlaps in frequency with fin whale songs and can cause a reduction of the communication space (CS) in this species (*Clark et al., 2009*; *Erbe et al., 2019*). Models estimate a reduction of CS by vessel noise of up to 80% for fin whales (*Cholewiak et al., 2018*; *Clark et al., 2009*). We ignore if fin whales use any anti-masking release mechanism when exposed to vessel noise (e.g. changing song frequencies), but if they do not, such reduction of CS could certainly disrupt communication and hinder the search of mates for reproduction, which ultimately would affect fin whale fitness.

Results from this and other studies suggest that male fin whales are in acoustic contact over vast areas and adjust their song properties to match those of conspecifics (*Leroy et al., 2018a*; *Oleson et al., 2014*; *Weirathmueller et al., 2017*). These acoustic communities culturally evolve more quickly and efficiently than genetic communities (*Hatch and Clark, 2004*) and should be considered in conservation strategies when delimiting stocks or populations. The acoustic habitat in which these songs evolved has shaped the acoustic properties and limits of variation of these signals. Understanding the cultural evolution of fin whale songs can inform us about the species' ability to adapt to the actual scenario of rapidly changing ocean soundscapes due to anthropogenic activities. These results also have implications for cue counting approaches, which use cue rates (e.g. notes per unit time) to convert density of sounds to density of animals (*Marques et al., 2013*). The temporal and spatial changes in fin whale song INIs found here affect cue rates and need to be considered to avoid bias in estimating densities using passive acoustic monitoring in this species. The unique large spatial

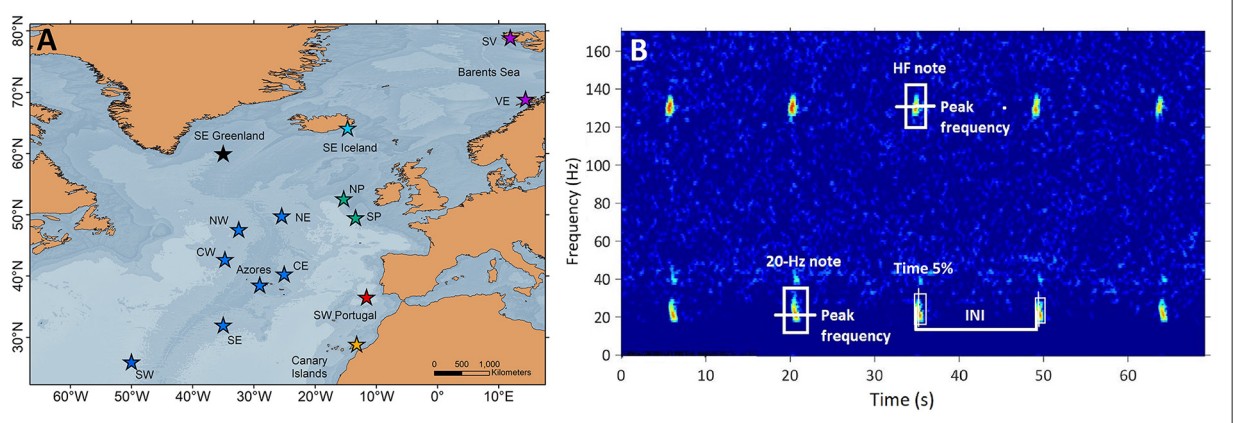

**Figure 5.** Sampling locations and fin whale song spectrogram. (**A**) Locations (stars) of acoustic recordings grouped in regions (colours in stars): SE Greenland (black), SE Iceland (turquoise), Celtic Sea (NP: North Porcupine and SP: South Porcupine; green), Oceanic Northeast Atlantic (NW, NE, CW, CE, Azores, SW and SE; blue), SW Portugal (red), Canary Islands (yellow) and Barents Sea (SV: Svalbard and VE: Vesterålen; purple). (**B**) Spectrogram (FFT sample duration 0.5 s, Hann window, 50% overlap) of a fin whale song showing the acoustic parameters analysed in this study (INIs and peak frequencies of the 20 Hz and HF note).

The online version of this article includes the following figure supplement(s) for figure 5:

**Figure supplement 1.** Sampling effort for the inter-note intervals (INIs) (left) and high frequency (HF) note (right) analysis for each location and singing season (between October and March).

scale over which fin whales communicate, although technologically challenging for researchers, opens interesting perspectives in the processes of animal acoustic communication.

## Materials and methods
### Sampling locations
Acoustic data were compiled from 15 locations in the central and northeast Atlantic Ocean, grouped into seven regions: SE Greenland, SE Iceland, Celtic Sea (North and South Porcupine), Oceanic Northeast Atlantic (ONA) (NE, NW, CE, CW, Azores, SE and SW), SW Portugal, Canary Islands and Barents Sea (Svalbard and Vesterålen; *Figure 5A*).

### Data collection
Recordings from 1999 to 2020 collected by different research groups with varied objectives were compiled and standardised. Not all regions were sampled in all years and time periods. Recordings were either continuous or duty-cycled with different sampling rates (*Table 1* and *Figure 5—figure supplement 1*). Ocean-Bottom Seismometers (OBS) were used in the Canary Islands (2014–2015) and SW Portugal (2007–2008). The hydrophone channel was selected for OBS recordings in the Canary Islands, while the seismometer channel (vertical component Z) was preferred for recordings from SW Portugal (2007–2008). Fixed autonomous recorders (AR) were used in the remaining regions (*Supplementary file 1b*).

### Song selection criteria
We focused the analyses on data collected between October and March (hereafter singing season), because fin whale song parameters show less variation during this period (*Hatch and Clark, 2004*) and seasonal variation was outside the scope of this study. All datasets were manually inspected to identify songs composed of 20 Hz notes (*Watkins et al., 1987*) or 20 Hz and HF note types (*Hatch and Clark, 2004*; *Figure 5B*), except for the Azores dataset, which had been analysed for another study using a Low Frequency Detection and Classification System (LFDCS) (*Baumgartner and Mussoline, 2011*) (procedures described in *Romagosa et al., 2020*). In all datasets, spectrograms of days with fin whale detections were manually analysed using Adobe Audition 3.0 software (Adobe Systems Incorporated, CA, USA) to select periods with good quality notes, based on: (a) clearly distinguishable song notes in the spectrogram (Signal to noise ratio - SNR >5 dB), (b) absence of masking from noise,

(c) presence of a single singer and, (d) occurrence of at least 10 notes organized in a series. The last criterion could not be applied for recordings with small duty cycles (SW Portugal 2015–2016, Azores 2008–2011 and the Celtic Sea) (*Table 1*); nevertheless, regularly spaced notes could still be identified as part of songs and were used for these sites. SNR was estimated for all selected 20 Hz notes by using the Inband Power measurement in Raven Pro 1.5 software (Cornell Lab of Ornithology, Ithaca, NY, USA) (*Supplementary file 1c*). For each selected note (Signal), a companion selection (Noise) was created and the Inband Power measured. Then we estimated SNR of each note by using the following formula (*Charif et al., 2010*):

$$SNR = \frac{Signal\ Inband\ Power - Noise\ Inband\ Power}{Noise\ Inband\ Power}$$

## Song sampling

Selected days with detections were non-consecutive to minimize the likelihood of sampling the same individual multiple times. The number of sampled days varied depending on the quality of fin whale songs found in the recordings. The average number of days sampled per singing season was 11.4 days, and the average number of notes analysed per song was 102 (*Figure 5—figure supplement 1*). Recordings from the Canary Islands, SW Portugal (2007–2008), and ONA regions, except for the Azores, were excluded from the analysis of the HF note, because sampling rates were too low to enable detection of the HF note frequency (~130 Hz) (*Hatch and Clark, 2004*; *Table 1*).

## Measurement of song parameters: INIs and peak frequencies

Selected days with good quality notes (see 'Song selection criteria' section) were fed into a band-limited energy detector in Raven Pro 1.5 software (Cornell Lab of Ornithology, Ithaca, NY, USA) that automatically selected all 20 Hz and HF notes in the spectrogram. All selections were checked manually by the same analyst to ensure that notes were well imbedded in the selection square. Spectrogram characteristics were adjusted to visualise all data with the same frequency and time resolution (1.25 Hz and 0.4 s). For each selected note, the software measured Begin and End time, Time of the 5% cumulative energy (Time 5%), Peak frequency and Inband Power (*Supplementary file 1c*). INIs were calculated by subtracting the time (Time 5%) difference between consecutive 20 Hz notes (*Širović et al., 2017*; *Watkins et al., 1987*; *Figure 5B*). This measurement calculates the point in time dividing the selection into two-time intervals containing 5% and 95% of the energy. Peak frequencies were measured for 20 Hz and HF notes and represent the value at which the maximum energy in the signal occurs. It is considered a robust measurement since it is based on the energy within the selection and not the time and frequency boundaries of the selection (*Charif et al., 2010*). Only one sequence of notes or song fragment (hereafter referred as song) was analysed per day in each location. If multiple songs were found in one day, the one with the highest SNR was selected. For each song, we calculated the mean and standard deviation of INIs and of peak frequencies of the 20 Hz and HF notes.

## Analysis of fin whale song INIs and note frequencies

Temporal patterns in song INIs were investigated by plotting mean song INIs and standard deviations of all regions into chronological order. Due to the identification of two song INIs during the first period of data (1999–2005), belonging to the ONA region, we also calculated the percentage of each song INI per singing season in this dataset. Specifically, the SE location of the ONA region, which had the longest time series (1999–2005), was used to investigate changes in song INI percentages over this period. The other locations of the ONA region had data only for the singing season of 2002/2003 and were used to investigate the spatial patterns in song INIs across six locations (NE, NW, CE, CW, SE and SW) (*Figure 5A* and *Figure 5—figure supplement 1*).

After this first period, data from all regions were plotted in chronological order to investigate how song parameters varied over time. A linear regression model was fit to each response variable (INIs and peak frequencies of the 20 Hz and HF notes) using a Gaussian distribution and year as the explanatory variable. Model assumptions were verified by plotting residuals versus fitted values and residual QQ plots to check for homogeneity of variance and normality (*Figure 3—figure supplement 1*). Measurements of 20 Hz peak frequencies were greatly affected by the recording equipment (*Supplementary file 1d* and *Figure 3—figure supplement 2*). For this reason, only data from the Ecological

Acoustic Recorders (EARs) (*Lammers et al., 2008*), which sampled the longest period (2008–2020) (*Table 1* and *Figure 5—figure supplement 1*), were used to explore temporal variations in the peak frequencies of the 20 Hz note. All statistical analyses were performed in R (v. 4.0.2) (*R Core team, 2020*).

## Regional comparison of song parameters

Given inter-annual variations in fin whale song parameters (*Delarue et al., 2009*; *Širović et al., 2017*; *Weirathmueller et al., 2017*), only songs recorded within the same singing season were used to compare song parameters among regions. Histograms were built for each singing season to investigate differences in the distribution of INIs and peak frequencies of the HF note per region sampled.

## Acknowledgements

Research was supported by the Portuguese Science & Technology Foundation (FCT), the Azorean Science & Technology Fund (FRCT) and the EC through research projects TRACE-PTDC/ MAR/74071/2006, MAPCET-M2.1.2/F/012/2011, FCT-Exploratory-IF/00943/2013 /CP1199/CT0001, AWARENESS-PTDC/BIA-BMA/30514/2017, co-funded by FEDER, COMPETE, QREN, POPH, ESF, Lisbon and Azores Regional Operational Programme, Portuguese Ministry for Science and Education. This work also recieved national funds through the FCT – Foundation for Science and Technology, I.P., under the project UIDB/05634/2020 and UIDP/05634/2020 and through the Regional Government of the Azores through the initiative to support the Research Centers of the University of the Azores and through the project M1.1.A/REEQ.CIENTÍFICO UI&D/2021/010. OKEANOS R&D Centre is supported by FCT through the strategic fund (UIDB/05634/2020). Canary Island data was provided by the Institut de Ciències del Mar under the 'Severo Ochoa Centre of Excellence' accreditation (CEX2019-000928-S). Data used from the ONA region is a NOAA-PMEL contribution number 5326. The Celtic Sea data belongs to the ObSERVE Acoustic project, initiated and funded by the Department of Communications, Climate Action and Environment in partnership with the Department of Culture, Heritage and the Gaeltacht under Ireland's ObSERVE Programme. Data collection in the Barents Sea region was made by the Italian CNR under the Arctic Field Grant project KUAM (235878/ E10), funded by the Norwegian Research Council through the Svalbard Science Forum, and under the Project Calving SEIS (244196/E10) funded by the Norwegian Research Council. Vesterålen data was provided by the LoVe Ocean Observatory project, led by the Institute of Marine Research and funded by the Norwegian Research Council and Equinor. Iceland data was collected under Velux Fonden and Knud Højgårds Fond funding. MR was supported by a DRCT doctoral grant (M3.1.a/F/028/2015). IC was supported by the FCT-IP Project UIDP/05634/2020. AP was supported by project AWARE-NESS 'PTDC/BIABMA/30514/2017' and 'UIDB/50019/2020–IDL'. TAM by CEAUL (funded by FCT - Fundação para a Ciência e a Tecnologia, Portugal, through the project UIDB/00006/2020) and the LMR ACCURATE project (contract no. N3943019C2176). R.P. was supported by an FCT grant (SFRH/ BPD/108007/2015). M.A.S. was funded by FCT (IF/00943/2013), EC (SUMMER H2020- EU.3.2.3.1, GA 817806) and the Operational Program AZORES 2020, through the Fund 01–0145-FEDER-000140 'MarAZ Researchers: Consolidate a body of researchers in Marine Sciences in the Azores' of the European Union. We are grateful to Marc Lammers, for providing the EARs and technical support, and to Sérgio Gomes, Norberto Serpa and all skilled skippers and crew that participated in the preparation and deployment of the EARs at DOP/IMAR and all other instruments used in this study. We also thank Dr. Christopher W Clark and the other two reviewers for significantly improving the original manuscript.

## Additional information

### Funding

| Funder | Grant reference number | Author |
| --- | --- | --- |
| Fundação para a Ciência e a Tecnologia | TRACE-PTDC/ MAR/74071/2006 | Rui Prieto Mónica A Silva |

| Funder | Grant reference number | Author |
|---|---|---|
| Fundo Regional para a Ciência e Tecnologia | MAPCET-M2.1.2/F/012/2011 | Rui Prieto<br>Mónica A Silva |
| Fundação para a Ciência e a Tecnologia | FCT-Exploratory-IF/00943/2013/CP1199/CT0001 | Rui Prieto<br>Mónica A Silva |
| Fundação para a Ciência e a Tecnologia | AWARENESS-PTDC/BIA-BMA/30514/2017 | Miriam Romagosa<br>Irma Cascão<br>Mónica A Silva |
| Fundação para a Ciência e a Tecnologia | UIDB/05634/2020 | Miriam Romagosa<br>Irma Cascão<br>Rui Prieto<br>Mónica A Silva |
| Fundación Carmen y Severo Ochoa | CEX2019-000928-S | Arantza Ugalde |
| National Oceanic and Atmospheric Administration | 5326 | Robert Dziak |
| Department of Communications, Climate Action and Environment | ObSERVE Programme | Joanne O'Brien |
| Consiglio Nazionale delle Ricerche | KUAM (235878/E10) | Elena Papale<br>Giuseppa Buscaino |
| Norwegian Research Council | Calving SEIS (244196/ E10) | Elena Papale<br>Giuseppa Buscaino |
| Norwegian Research Council | | Sofia Aniceto |
| Velux Fonden | | Marianne Rasmussen |
| Knud Højgårds Fond | | Marianne Rasmussen |
| Fundo Regional para a Ciência e Tecnologia | M3.1.a/F/028/2015 | Miriam Romagosa |
| Fundação para a Ciência e a Tecnologia | UIDP/05634/2020 | Irma Cascão |
| Fundação para a Ciência e a Tecnologia | UIDB/50019/2020-IDL | Andreia Pereira |
| Fundação para a Ciência e a Tecnologia | UIDB/00006/2020 | Tiago A Marques |
| US Navy | Living Marine Resources N3943019C2176 | Tiago A Marques |
| Fundação para a Ciência e a Tecnologia | SFRH/BPD/108007/2015 | Rui Prieto |
| Fundação para a Ciência e a Tecnologia | IF/00943/2013 | Mónica A Silva |
| H2020 European Institute of Innovation and Technology | SUMMER H2020-EU.3.2.3.1 | Mónica A Silva |
| Operational Program AZORES 2020 | 01-0145-FEDER-000140 | Mónica A Silva |
| NOAA Pacific Marine Environmental Laboratory | 5025 | David K Mellinger |
| H2020 European Institute of Innovation and Technology | GA 817806 | Mónica A Silva |

| Funder | Grant reference number | Author |
|---|---|---|
| Fundação para a Ciência e a Tecnologia | UIDP/05634/2020 | Miriam Romagosa<br>Irma Cascão<br>Rui Prieto<br>Mónica A Silva |
| Governo Regional dos Açores | M1.1.A/REEQ.CIENTÍFICO UI&D/2021/010 | Miriam Romagosa<br>Irma Cascão<br>Rui Prieto<br>Mónica A Silva |

The funders had no role in study design, data collection and interpretation, or the decision to submit the work for publication.

## Author contributions
Miriam Romagosa, Conceptualization, Data curation, Formal analysis, Investigation, Visualization, Methodology, Writing – original draft; Sharon Nieukirk, Conceptualization, Resources, Data curation, Writing – review and editing; Irma Cascão, Robert Dziak, Joanne O'Brien, David K Mellinger, Andreia Pereira, Arantza Ugalde, Elena Papale, Sofia Aniceto, Marianne Rasmussen, Luis Matias, Rui Prieto, Resources, Writing – review and editing; Tiago A Marques, Supervision, Methodology, Writing – review and editing; Jean-Yves Royer, Giuseppa Buscaino, Resources; Mónica A Silva, Conceptualization, Resources, Supervision, Funding acquisition, Validation, Writing – original draft, Project administration, Writing – review and editing

## Author ORCIDs
Miriam Romagosa ⬤ http://orcid.org/0000-0003-2781-5528
Irma Cascão ⬤ http://orcid.org/0000-0001-6231-0483
David K Mellinger ⬤ https://orcid.org/0000-0002-5228-0513
Mónica A Silva ⬤ https://orcid.org/0000-0002-2683-309X

## Decision letter and Author response
Decision letter https://doi.org/10.7554/eLife.83750.sa1
Author response https://doi.org/10.7554/eLife.83750.sa2

---

# Additional files

## Supplementary files
• Supplementary file 1. Supplementary information on sampliing of fin whale songs and recording equipment. (a) Table showing information of each song analysed in this study: date, time of the first (Time first note) and last note analysed (Time last note), duration and number of INIs measured (Num. INIs). (b). Position, recording equipment and approximate (Approx.) depth for each location sampled. (c). Description of fin whale song measurements made with Raven Pro 1.4 software (*Charif et al., 2010*). (d) The effect of recording equipment on fin whale song parameters When comparing data from multiple sensors an obvious question is whether the results might be dependent on the specific sensors considered. To investigate the influence of the acoustic recorder type, OBS or ARs, on the song parameters, we analysed the same song fragment, consisting of 209 notes, recorded by the hydrophone channel of an OBS and an AR, specifically an Ecological Acoustic Recorder (EAR) (*Lammers et al., 2008*). The two instruments were deployed at ~6 km from each other in the Azores region in spring of 2019. Measurements of INIs and 20 Hz peak frequencies of songs recorded by each instrument were compared using a non-parametric paired samples Wilcoxon Test. Differences in HF note peak frequencies could not be tested because of limitations in the sampling rate of the OBS. Results showed that median INIs measured from OBS (16.52) and EARs (16.42) were not significantly different (*P*-value = 0.47) but median peak frequencies of the 20 Hz note (median OBS = 23.4; EARs = 21.1) were (*P*-value <0.001). Thus, the use of different recorders did not affect INI measurements but influenced measurements of 20 Hz peak frequencies. The effect of the distance to the source in the analysed fin whale song parameters is included in this study, given that these two recorders were positioned at different distances to the singer. Peak frequencies of the 20 Hz note showed a great variability between equipment types (Fig. S3), which hindered the identification of soft trends (i.e., low changing rate). For this reason, only data from the EARs, the longest dataset (2008 – 2020), were used to study temporal variations of the peak frequencies of the 20 Hz note. All

statistical analyses were performed using the software R (v. 4.0.2) (*R Core team, 2020*).

• MDAR checklist

### Data availability

All datasets and R scripts used in this study have been deposited in the Dryad Digital Repository.

The following dataset was generated:

| Author(s) | Year | Dataset title | Dataset URL | Database and Identifier |
|---|---|---|---|---|
| Romagosa M, Sharon N, Cascâo I, Marques T, Dziak R, Royer J, O'Brien J, Mellinger D, Pereira A, Ugalde A, Papale E, Aniceto S, Buscaino G, Rasmussen M, Matias L, Prieto R, Silva MA | 2023 | Data and R code from: Fin whale song evolution in the North Atlantic | https://dx.doi.org/10.5061/dryad.f1vhhmh13 | Dryad Digital Repository, 10.5061/dryad.f1vhhmh13 |

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
