## [Editor Report]

This study is a valuable contribution to our understanding of vocal variation in acoustic displays of male baleen whales, part of a developing story about cultural change in songs in species other than the relatively well studied humpback whales. The authors present solid evidence of changes at various timescales in 20-Hz song note intervals and call center frequency over decadal time scales and large spatial scales.

---

## [Decision Letter]

**Decision letter after peer review:**

Thank you for submitting your article "Fin whale singalong: evidence of song conformity" for consideration by *eLife*. Your article has been reviewed by 3 peer reviewers, one of whom is a member of our Board of Reviewing Editors, and the evaluation has been overseen by Christian Rutz as the Senior Editor. The following individuals involved in review of your submission have agreed to reveal their identity: Christopher W Clark (Reviewer #1).

Overall, our decision is that the manuscript is not suitable for publication in its current form, but we invite you to revise it following the feedback given below. The reviewers have discussed their reviews with one another, and the Reviewing Editor has drafted this to help you prepare a revised submission. The revisions we ask for are substantial, but hope you agree that they will improve the manuscript, and I encourage you to take them on board – failing to do so to the satisfaction of the reviewing team may lead to rejection.

Essential revisions:

1. Having looked at the data and the analysis, we are not convinced the main story here is about conformity. Isn't the picture more about the fact that there are continual gradual changes which raises the question as to how changes could keep going in the same direction once they start to hit physiological limits, and now, we have a possible answer – periodically they reset? This is a really useful piece of understanding, but it would be better if the paper centred on this as a principal question – the gradual changes were already known about in blues and to some degree in fins (e.g. Rankin /Stafford/Castellote Oleson etc. etc.), but now we see how some song features of a male fin whale singer population can remain within certain bounds and not infinitely (and impossibly) reduce or change song characteristics. Please revise in the light of these arguments.

2. We are not convinced the authors are correct in describing this change in rate as a change of song type. No definition of 'type' in fin whale song is given. Even though fin whale calls are evidently a male mating ground display, little is known about its function. Compared to humpback whales with their impressive repertoire of vocalizations, repeating themselves on the breeding grounds after some tens of minutes and therefore qualifying as a very slow 'song' similar to bird song, fin whale songs in the North Atlantic are composed from three simple note types, and they are only changing pitch and timing – it is not really a new type of song. Please revise in the light of these arguments.

3. The abstract states: "We also revealed gradual changes in INIs and note frequencies over more than a decade with all males adopting both rapid and gradual changes." The data are such that the authors could not determine if an individual singer adopted both rapid and gradual changes. A singer only contributed a data point to the overall collection of INI and frequency values. As such, if individual males could not be shown to actually adopt changes in INIs and note frequencies over more than a decade, how could all males be shown to do this? Please reflect on the nature of your data and how it relates to changes within and between individuals in the light of these arguments.

4. Please undertake a major rewrite of the introduction – currently it lacks depth and breadth – improving one of these should be sufficient – for depth, for example, what are the various motivations for understanding song variation in fin whales and in other taxa? L89-91 is not sufficient. Also, a discussion of genetic population structure in fin whales should be included – spatially this study relies on ICES ecoregions but the relevance of these for fin whales is unclear. The inclusion and treatment of avian song learning studies is simplistic and not accurate in places. Given the immense amount of multi-decadal, multi-species, impactful research on captive and free-ranging birds, the reduction of the level of understanding of avian song acquisition to "learning to vocalize through imitation" is both incorrect and unacceptable. At best, a few selected references to the most appropriate discoveries and synthesis should be cited, for example, Marler and Slabbekoorn, Nature's Music: The Science of Birdsong. Consider also Mennill's work on cultural transmission and consider also Otter et al. Curr Bio 2020.

5. It took reviewers some time reading this manuscript to figure out that the analyses were focussed on two different time periods – the ONA data showing the apparent transition from the 19s INI to the 12s INI in the early 2000s and then the broader scale data showing more gradual trends apparently paralleled across the study area. We think the presentation could go a lot further to making these 2 different analyses more clearly distinguished – one local showing abrupt change in one region and then one on a larger scale showing the gradual change over time.

6. The points about conformity on L49 could be better supported by citations – in fact, this is not a well-known definition of conformity – the cultural evolution literature is replete with them – look to papers by e.g. Morgan, Whiten, Laland – and the learning of nearby songs over distant ones can be explained simply by the physics of transmission loss under the reasonable assumption that the clearer you hear a song the more likely you are to learn it. It still leads to interesting patterns of variation and does not argue against social learning still underlying song variation, but we are less convinced by this conformity argument.

7. In places it sounds like the authors are writing from a slightly defensive posture as if they feel bad somehow that fin whale songs are less 'complex' than, say, humpbacks. We encourage the authors to be specific about what is meant by complexity (e.g. number of different song unit types, or number of phrase types in a song), reflect on whether the relationship between complexity and interpretability is as straightforward as they suggest, and finally reflect on why they apparently buy into the idea that 'complexity=interesting' assumption that they feel the need to defend their study against, and revise accordingly.

8. L75 "studies on fin whale song changes have been merely descriptive"

- how is this study not also descriptive? Reviewers disliked the descriptor 'merely' for descriptive studies – try posing a hypothesis without leaning on some form of description of your study system! This might be better focussed on some strengths of this study like temporal and spatial coverage.

9. L79-80 the phrase 'unknown mechanisms of vocal behaviour' is a bit mysterious please be more explicit in your meaning here.

10. L77 this para was vague on the motivations of the study – 'report the dynamics' is a kind of place-holder from which it is hard to predict the actual analyses that were run. Why did you suspect there might be such dynamics? What aspects did you think would be subject to geographic and temporal variation, and why?

11. The lack of proper inclusion and interpretation of previous studies documenting seasonal and multi-annual changes in fin whale INI needs to be addressed (e.g., Delarue et al. 2009, Morano et al. 2012). For example, Morano et al. 2012 shows seasonal INI variations and fluctuation between 10s and 16s.

12. L95 – Figure 1A shows a map, not changes in frequency?

13. L95 – Two 'songs' are described here as if the reader should know what they are but they haven't been defined yet. Perhaps explain here what you mean by the '19s-INI' song and the rules you used to differentiate between the songs. Reviewers would ask you to start with a strictly factual description of the results before adding interpretations such as 'we named these two different INI periods as two different song types'. We are not sure that these are different songs – there are definitely different rhythms, but actually the units and their order are exactly the same. So they might be better described as distinct 'tempos' or 'rhythms' rather than completely different songs?

14. L97 why is only the SE ONA region presented in such detail – what happens at the other recorders concurrently? And why is only 2002/2003 presented for the entire ONA region?

15. L99-100 – define a 'hybrid song' and how they were identified – would also be interesting to know how the different INIs were distributed over the sequence – first one then the other, or more mixed?

16. The text mentions acquisition duty-cycle with a reference to Table S1, but that table does not list the start date-time and end date-time of the sampling. If one goes to Figure S1 showing a color-coded listing of recording effort, the time resolution is in months, and data in this chart is not linked to sample rate or duty-cycle. A reader must be given all the necessary information by which to assess data quality, validity of analysis results and reliability of interpretation.

17. Along similar lines: L300-301 – how was the 'lowest SNR' channel selected – from measurements, estimations, sampling? How was SNR measured? Important details such as duty cycle and sampling rate are in a supplementary table, but I think they are important enough to earn their place in the main text.

18. L177 this sentence confused the reviewers – it seems to contradict itself?

19. L241-242 – isn't the adoption of a new song actually evidence *against* conformity? Since in a conformist scenario, the new song has to get common enough to be the majority option?

20. L259-260 – We urge the authors to think more critically about the suggestion that humpback song could function as a sonar (for example, considering the long-standing arguments of Au and Cato against this notion) but it is worth noting that if any baleen whale song were to function in some kind of sonar way then they would be better adapted in terms of their regular structure – however, the frequencies being so low really do make it an unlikely function.

21. Text concludes that there was a "complete turnover" of INI. This is a rather strong conclusion given the spatial and temporal sparseness of the data. Is this the right form of words to describe the observations?

22. The time-warping used in Figure 2A, Figure 3A, B and C is a bit misleading. Excluding time gaps in available data distorts the time series and could obscure an actual structure (tendency.) We suggest remaking Figure 2A such that the time scale is uniformly linear, not dilated as in a form of dynamic time warping. As presented, it is difficult to interpret. For example, what does the 2001-2002 period include, October 2001 through March 2002? But it could be that all these samples are from Nov 2001 through Jan 2002. Maybe monthly tic marks would help? How about horizontal lines at each of the integer INI values? This might help the reader see slight increases in INI slope over time? Figure S1 is an incomplete synthesis of the data set. It would be valuable to have more and better represented details. We suggest rearranging this figure so that the time months are based on the singing season, not the human calendar. This would have the left-to-right 6six month period go from Oct through March, thereby representing the behavior from the species' behavioral biology perspective rather than a human calendar. This simplifies the representation of when and where each of the possible 20 singing seasons is sampled. It would also be beneficial to clearly show how much each of the sites contributed to the INI and frequency samples.

23. L309-311 – given that two different methodologies were used (manual and LFDCS), was there any test of the two systems on the same recordings to check they are yield comparable results?

24. L315 specify how long a 'long series' was.

25. The text and figures are not always consistent. For example, the text refers to the E region as the "Bay of Biscay and Iberian Coast", when it contained only a single sensor deployed off the southwest of the Iberian Peninsula. This contrast between the realities of where and when acoustic samples were collected and how the aggregated results are spatially or temporally interpreted or labelled persists throughout the manuscript.

26. Supplementary material needs a table listing the details of the acoustic data recording effort (e.g., from date-time, to date-time) and the acoustic analysis effort (e.g., total recording hours per deployment, for each deployment at each site list the proportion of the total hours relative to the grand total of recording hours, INI-song dates, song time of day start, duration, number of notes.) For example in Figure S1, in January of 2003 for the SE site in the Oceanic Northeast Atlantic region, a song from each of four different days was analyzed yielding a total of 556 INI values for that month. Data from this SE site is the dominant source of recordings in this study. Section 1 in the Supplemental Material is headed in the right direction of addressing sampling aliasing influences on outcome, and this type of effort needs to be included given the large differences in sampling effort/site.

27. We could not find any details on analysis parameters except for the spectrogram figure in Figure 1, where it gives 1024-point FFT, Hann window, 50% overlap. There was no mention of time and frequency resolutions. E.g., if using 1024-point FFT, Hann window, 50% overlap, then the time resolution is 0.26 s and frequency is 1.95 Hz. These resolutions determine the ability to measure and track changes in INI and note peak frequencies. We may have missed seeing a listing of the measurements collected using Raven, but if they weren't there, then they should be included.

28. L335-337 – the measurement of INIs is insufficiently explained – the figure implies this was a separate measurement from the Raven selection boxes, yet there are robust measurement methods in Raven for measuring onsets based on energy content over time (e.g the 5% cumulative energy threshold).

29. L350 song types were known prior to this study? Otherwise, this is a result in the wrong place…

30. L360 – 'except for the Barent's Sea which showed different INIs' – this is a little problematic since it implies you excluded data not for any a-priori reason but because it didn't fit with your preferred model – it is obvious why such a process would not be conducive to rigour. If there is reason to believe the whales from that region belong to an entirely different population and thereby establish an a-priori reason for their exclusion then that should be presented and justified, but excluding data from analyses after results are known isn't a robust way to go about things.

31. The degree to which this is evidence of vocal learning may be a bit overplayed. Definitely there is change, but it is tricky to compare this to e.g. experimental demonstrations. For example, age-related changes in a changing post-whaling demographic scenario should at least be considered?

32. The document contains various grammatical and punctuation errors, and there are several errors in the reference section. Please proof-read the manuscript thoroughly.

*Reviewer #1 (Recommendations for the authors):*

This is a valuable piece of research and represents a tremendous amount of work. Overall, it is well written and well organized. The extent of these comments reflects the extent of my keen interest in the topic as well wanting to strengthen the manuscript's scientific integrity. With some further effort, it is certainly worthy of publication somewhere, but I have not been asked to render a decision as to where it might best be published.

These following comments are not exhaustive. They are representative. The more I continued to read, reread, and focus on certain aspects of this very interesting and well written paper, the more worried I became about of the influences on sampling aliases on the results and the interpretations of those results. For example, the text mentions acquisition duty-cycle with a reference to Table S1, but that table does not list the start date-time and end date-time of the sampling. If one goes to Figure S1 showing a color-coded listing of recording effort, the time resolution is in months, and data in this chart is not linked to sample rate or duty-cycle. A reader must be given all the necessary information by which to assess data quality, validity of analysis results and reliability of interpretation.

The inclusion and treatment of avian song learning studies is remarkably simplistic and certainly not scholarly accurate. Given the immense amount of multi-decadal, multi-species, impactful research on captive and free-ranging birds, the reduction of the level of understanding of avian song acquisition to "learning to vocalize through imitation" is both incorrect and unacceptable. At best, a few selected references to the most appropriate discoveries and synthesis should be cited, for example, Marler and Slabbekoorn, Nature's Music: The Science of Birdsong.

The lack of proper inclusion and interpretation of previous studies documenting seasonal and multi-annual changes in fin whale INI is perplexing (e.g., Delarue et al. 2009, Morano et al. 2012). For example, Morano et al. 2012 shows seasonal INI variations and fluctuation between 10s and 16s.

Text concludes that there was a "complete turnover" of INI. This is a rather strong conclusion given the spatial and temporal sparseness of the data. Is this the right form of words to describe the observations?

I've already made comments about the sampling bias resulting from the high variability in sampling dates and durations. I found time-warping used in Figure 2A, Figure 3A, B and C inappropriately misleading. Excluding time gaps in available data distorts the time series and could obscure an actual structure (tendency.) I don't doubt that the statistical evaluations in the Supplemental materials are appropriate, but Figure S1 is an incomplete synthesis of the data set. It would be valuable to have more and better represented details. I suggest rearranging this figure so that the time months are based on the singing season, not the human calendar. This would have the left-to-right six month period go from Oct through March, thereby representing the behavior from the species' behavioral biology perspective rather than a human calendar. This simplifies the representation of when and where each of the possible 20 singing season is sampled. It would also be beneficial to clearly show how much each of the sites contributed to the INI and frequency samples.

There is a tendency to slightly, and likely inadvertently, misstate or overstate a result. For example, in the Abstract: "We also revealed gradual changes in INIs and note frequencies over more than a decade with all males adopting both rapid and gradual changes." The data are such that the authors could not determine if an individual singer adopted both rapid and gradual changes. A singer only contributed a data point to the overall collection of INI and frequency values. As such, if individual males could not be shown to actually adopt changes in INIs and note frequencies over more than a decade, how could all males be shown to do this? The unit of analysis needs to be very clearly stated and adhered to throughout the manuscript. The observation of rapid and gradual changes is derived by aggregating INI results from all singers (N = sum of all monthly song samples?) and is not based on the behavior of single individual singers.

The text and figures are not always consistent. For example, the text refers to the E region as the "Bay of Biscay and Iberian Coast", when it contained only a single sensor deployed off the southwest of the Iberian Peninsula. This contrast between the realities of where and when acoustic samples were collected and how the aggregated results are spatially or temporally interpreted or labelled persists throughout the manuscript. Why are ICES ecoregions needed or useful?

Suggest remaking Figure 2A such that the time scale is uniformly linear, not dilated as in a form of dynamic time warping. As presented, it is difficult to interpret. For example, what does the 2001-2002 period include, October 2001 through March 2002? But it could be that all these samples are from Nov 2001 through Jan 2002. Maybe monthly tic marks would help? How about horizontal lines at each of the integer INI values? This might help the reader see slight increases in INI slope over time?

Supplementary material needs a table listing the details of the acoustic data recording effort (e.g., from date-time, to date-time) and the acoustic analysis effort (e.g., total recording hours per deployment, for each deployment at each site list the proportion of the total hours relative to the grand total of recording hours, INI-song dates, song time of day start, duration, number of notes.) If I read Figure S1 correctly, for example, in January of 2003 for the SE site in the Oceanic Northeast Atlantic region, a song from each of four different days was analyzed yielding a total of 556 INI values for that month. Data from this SE site is the dominant source of recordings in this study. Section 1. In Supplemental material is headed in the right direction of addressing sampling aliasing influences on outcome, and this type of effort needs to be included given the large differences in sampling effort/site.

I could not find any details on analysis parameters except for the spectrogram figure in Figure 1, where it gives 1024-point FFT, Hann window, 50% overlap. There was no mention of time and frequency resolutions. E.g., if using 1024-point FFT, Hann window, 50% overlap, then the time resolution is 0.26 s and frequency is 1.95 Hz. These resolutions determine the ability to measure and track changes in INI and note peak frequencies. I must have missed seeing a listing of the measurements collected using Raven, but if they weren't there, then they should be included.

In Figure 1B, it appears that the time of occurrence of a 20-Hz note is taken around the time at which the analyst visually determined the "begin time" of the note. This is not a reliable means of determining the time of occurrence of an acoustic event. The same issue regarding frequency resolution applies (see Fristrup and Watkins 1993, and Raven Pro User's Manual). I don't think this necessarily will make a huge difference in the basic message in the manuscript, but it does impact the resolution and uncertainty in the two major measurements in this manuscript.

The document contains various grammatical and punctuation errors, and there are several errors in the reference section. If I've misinterpreted the text, figures, or supplemental materials, these are certainly unintentional, and I extend my apologies. The paper was very interesting, and I have tried my best to understand it from a variety of perspectives, while being forthright about what I feel must be corrected.

*Reviewer #2 (Recommendations for the authors):*

It took me some time reading this manuscript to figure out that the analyses were focussed on two different time periods – the ONA data showing the apparent transition from the 19s INI to the 12s INI in the early 2000s and then the broader scale data showing more gradual trends apparently paralleled across the study area. I think the presentation could go a lot further to making these 2 different analyses more clearly distinguished – one local showing abrupt change in one region and then one on a larger scale showing the gradual change over time.

Having looked at the data and the analysis I am not convinced the main story here is about conformity. Isn't the picture more about the fact that there are continual gradual changes which initially raised the query as how this could keep going and now we have an answer – periodically they reset. This is a really useful piece of understanding but it would be better in my view if the paper centred this as a principal question – the gradual changes were already known about in blues and to some degree in fins (e.g. Rankin/Castellote) but now we see how the population can stay within certain bounds and not infinitely (and impossibly) reduce or change song characters. Why this happens – gradual change with periodic 'resets' – is similar but not precisely parallel to humpbacks – mainly because I don't think these are different 'songs' – the units are the same, the structure is the same, just some quantitative aspects of the song change. This doesn't diminish the importance for me, but just is more accurate – I think it would be a mistake to too easily reach for the humpback comparison – the differences are fascinating and worthy of further study – why such simple songs compared to the humpback acoustic peacock tail?? Is it related to functionality, or to different population structure and breeding patterns? And why the changes? I agree such directional changes make conformity a plausible process, and this is especially the case if whales changed their individual songs at the 2000 'reset' – fascinating to reflect on how that might be instigated or triggered, but also important to consider whether range shifts/population movements and/or demographic changes might mean it's actually different individuals or ontogenetic changes.

Introduction:

In general the introduction lacks depth and breadth – improving one of these should be sufficient – for depth, for example, what are the various motivations for understanding song variation in fin whales and in other taxa – L89-91 is not sufficient. Also, a discussion genetic population structure in fin whales should be included – spatially this study relies on ICES ecoregions but the relevance of these for fin whales is unclear.

48 used in a different context or sequence. When individuals are more likely to share song

49 variants with nearby individuals than with more distant ones, we talk about conformity

- this could be better supported by citations – in fact, this is not a definition of conformity I have met before (the cultural evolution literature is replete with them) – look to papers by e.g. Morgan, Whiten, Laland – and to me the learning of nearby songs over distant ones can be explained simply by the physics of transmission loss under the reasonable assumption that the clearer you hear a song the more likely you are to learn it. It still leads to interesting patterns of variation and does not argue against social learning still underlying song variation, but I am less convinced by this conformity argument.

62 has focused on complex songs and vocal learning of rhythm

67 whales or songbirds, offers an easier-to-interpret scenario

– I don't really get these points – it sounds like the authors are writing from a slightly defensive posture as if they feel bad somehow that fin whale songs are less 'complex' than say humpbacks. I would encourage the authors to be specific about what is meant by complexity (e.g. number of different song unit types, or number of phrase types in a song), reflect on whether the relationship between complexity and interpretability is as straightforward as they suggest, and finally reflect on why they apparently buy into the idea that 'complexity=interesting' assumption that they feel the need to defend their study against.

75 far, studies on fin whale song changes have been merely descriptive

– how is this study not also descriptive? I strongly dislike the descriptor 'merely' for descriptive studies – try posing a hypothesis without leaning on some form of description of your study system! This might be better focussed on some strengths of this study like temporal and spatial coverage.

79 scope of cultural song evolution, which provide a unique opportunity to investigate unknown

80 mechanisms of vocal behaviour in this species.

– the phrase 'unknown mechanisms of vocal behaviour' is a bit mysterious to me – could the authors be more explicit in their meaning here.

L77 this para more like an abstract? I found it vague on the motivations of the study –

'report the dynamics' is a kind of place-holder from which it is hard to predict the actual analyses that were run. Why did you suspect there might be such dynamics? What aspects did you think would be subject to geographic and temporal variation, and why?

Methods:

L300-301 – how was the 'lowest SNR' channel selected – from measurements, estimations, sampling? How was SNR measured? Important details such as duty cycle and sampling rate are in a supplementary table but I think they are important enough to earn their place in the main text.

L309-311 – given that two different methodologies were used (manual and LFDCS) was there any test of the two systems on the same recordings to check they are comparable.

L315 specify how long a ‘long series’ was.

L333-334 – specifically what time and frequency resolutions? This is important to put measured changes in context, especially the very subtle frequency ones.

L335-337 – the measurement of INIs is insufficiently explained – the figure implies this was a separate measurement from the Raven selection boxes, yet there are robust measurement methods in Raven for measuring onsets based on energy content over time (e.g the 5% cumulative energy threshold).

L340 what was the minimum number of notes or units required for–a sequence to be included in the analysis?

L342 was ‘highest SNR’ measured or estimated?

L345-346 ‘the only area with recordings during the song transition’.

I struggled with this as part of the methods – surely the song transition is a *result* of the study, not something that should have been used in making analysis decisions – the implied circularity ('we only looked here because here was the only place that had what we were looking for') is rather seriously undermining. It would be better to set out some clearer general objectives in the introduction and then make it clear that ONA specific analyses were conducted only *after* a result of interest was found in that location – this is fundamental to not ending up in the garden of forking paths where data and outcomes determine analysis decisions, rather than pre-specified hypotheses.

L350 song types were known prior to this study? Otherwise this is a again a result in the wrong place…

L360 – 'except for the Barent's Sea which showed different INIs' – this is a little problematic since it implies you excluded data not for any a-priori reason but because it didn't fit with your preferred model – I think it is obvious why such a process would not be conducive to rigour. If there is reason to believe the whales from that region belong to an entirely different population and thereby establish an a-priori reason for their exclusion then that should be presented and justified, but excluding data from analyses after results are known isn't the way to go about things.

L361 – explain more why frequency measurements were greatly affected here.

Figure S1 is quite confusing – can this be re-organised as a gant chart with time on the x axis and site on the y?

Figure S2 has some title glitches.

Results:

L95 – Figure 1A shows a map, not changes in frequency?

L95 – two 'songs' are described here as if the reader should know what they are but they haven't been defined yet. Perhaps explain here what you mean by the '19s-INI' song and the rules you used to differentiate between the songs. I think you need to start with a strictly factual description of the results before adding interpretations such as 'we named these two different INI periods as two different song types'. I am not sure I buy that these are different songs – there are definitely different rhythms, but actually the units and their order are exactly the same. So they might be better described as distinct 'tempos' or 'rhythms' rather than completely different songs?

L97 why is only the SE ONA region presented in such detail – what happens at the other recorders concurrently? And why is only 2002/2003 presented for the entire ONA region?

L98 'except FOR'.

L98 'isolated account' – you mean from the Barent's Sea – I think this deserves more than the dismissive treatment here – you are confounding location and time and should perhaps give a little more attention to this observation.

L99-100 – define a 'hybrid song' and how they were identified – would also be interesting to know how the different INIs were distributed over the sequence – first one then the other, or more mixed?

Discussion:

I think perhaps the degree to which this is evidence of vocal learning may be a bit overplayed. Definitely there is change, but it is tricky to compare this to e.g. experimental demonstrations. For example, age-related changes in a changing post-whaling demographic scenario should at least be considered?

L177 this sentence confused me – it seems to contradict itself?

L241-242 – isn't the adoption of a new song actually evidence *against* conformity? Since in a conformist scenario, the new song has to get common enough to be the majority option?

L259-260 – I urge the authors to think more critically about the suggestion that humpback song could function as a sonar (for example, considering the long-standing arguments of Au and Cato against this notion) but it is worth noting that if any baleen whale song were to function in some kind of sonar way then they would be better adapted in terms of their regular structure – however, the frequencies being so low really do make it an unlikely function.

*Reviewer #3 (Recommendations for the authors):*

I suggest the authors downplay the rather seeked take-home messages re. song learning and song conformity; just reporting the actual results, interesting as they are, should suffice, without making claims of learning et cetera.

---

## [Author Response]

Essential revisions:1. Having looked at the data and the analysis, we are not convinced the main story here is about conformity. Isn't the picture more about the fact that there are continual gradual changes which raises the question as to how changes could keep going in the same direction once they start to hit physiological limits, and now, we have a possible answer – periodically they reset? This is a really useful piece of understanding, but it would be better if the paper centred on this as a principal question – the gradual changes were already known about in blues and to some degree in fins (e.g. Rankin /Stafford/Castellote Oleson etc. etc.), but now we see how some song features of a male fin whale singer population can remain within certain bounds and not infinitely (and impossibly) reduce or change song characteristics. Please revise in the light of these arguments.

We have reviewed the manuscript according to the reviewers’ perspective that conformity is not the main story that can better reflect our data and analysis. First, we have changed the title from “Fin whale singalong: evidence of song conformity” to “Fin whale song evolution in the North Atlantic”. Then, we have made major changes in the introduction and discussion so that the manuscript does not focus on conformity but on cultural transmission, song evolution and the limits of song variation. We believe that fin whales show enough evidence of song cultural transmission (from this and other studies), and this is why we decided to maintain this topic in the introduction and discussion.

2. We are not convinced the authors are correct in describing this change in rate as a change of song type. No definition of 'type' in fin whale song is given. Even though fin whale calls are evidently a male mating ground display, little is known about its function. Compared to humpback whales with their impressive repertoire of vocalizations, repeating themselves on the breeding grounds after some tens of minutes and therefore qualifying as a very slow 'song' similar to bird song, fin whale songs in the North Atlantic are composed from three simple note types, and they are only changing pitch and timing – it is not really a new type of song. Please revise in the light of these arguments.

The reviewers raise an interesting point here and because we cannot prove that the different INI types here belong to different song types, we have replaced the term song type to song INI type. In the discussion, we hypothesise that these INI types may belong to different song types based on previous evidence that INIs are regionally distinct and could be considered different song types that differentiate populations.

3. The abstract states: "We also revealed gradual changes in INIs and note frequencies over more than a decade with all males adopting both rapid and gradual changes." The data are such that the authors could not determine if an individual singer adopted both rapid and gradual changes. A singer only contributed a data point to the overall collection of INI and frequency values. As such, if individual males could not be shown to actually adopt changes in INIs and note frequencies over more than a decade, how could all males be shown to do this? Please reflect on the nature of your data and how it relates to changes within and between individuals in the light of these arguments.

We acknowledge this sentence was not accurate or did not reflect the nature of our data. We have now removed the terms “all males” by just simply “sampled fin whale songs”, not specifying if all individuals or not adopt song changes.

4. Please undertake a major rewrite of the introduction – currently it lacks depth and breadth – improving one of these should be sufficient – for depth, for example, what are the various motivations for understanding song variation in fin whales and in other taxa? L89-91 is not sufficient. Also, a discussion of genetic population structure in fin whales should be included – spatially this study relies on ICES ecoregions but the relevance of these for fin whales is unclear. The inclusion and treatment of avian song learning studies is simplistic and not accurate in places. Given the immense amount of multi-decadal, multi-species, impactful research on captive and free-ranging birds, the reduction of the level of understanding of avian song acquisition to "learning to vocalize through imitation" is both incorrect and unacceptable. At best, a few selected references to the most appropriate discoveries and synthesis should be cited, for example, Marler and Slabbekoorn, Nature's Music: The Science of Birdsong. Consider also Mennill's work on cultural transmission and consider also Otter et al. Curr Bio 2020.

We have made a major rewrite of the introduction, so conformity is not the principal topic of our manuscript. We now focus on song evolution mechanisms, cultural transmission and the limits of song variation. To do so, we have reviewed a few examples of song evolution in birds, because it is the best studied taxa on the topic, and then we mentioned a few aspects of humpback whale song evolution to go straight to fin whales from there. We have further extended the fin whale paragraph on fin whale songs and added a genetic review from the North Atlantic and how this relates to the acoustic properties of songs in this species. This leads to the motivations of investigating song variations in animals and how can we improve this knowledge. Regarding the area divisions based on the ICES ecoregions, we have adopted our own divisions based on practicality. For example, we have kept the ONA region in the Mid-North Atlantic Ocean facilitate the explanation of song changes in the first sampled period. In the discussion we have added more accurate information on birds’ vocal learning, following the reviewers’ recommendations.

5. It took reviewers some time reading this manuscript to figure out that the analyses were focussed on two different time periods – the ONA data showing the apparent transition from the 19s INI to the 12s INI in the early 2000s and then the broader scale data showing more gradual trends apparently paralleled across the study area. We think the presentation could go a lot further to making these 2 different analyses more clearly distinguished – one local showing abrupt change in one region and then one on a larger scale showing the gradual change over time.

We acknowledge that the structure of the results was confusing, and the different analyses and time periods were not clearly explained. We have changed the titles of the Results section and figures to be clearer and more direct in terms of process, region and sampling period analysed. We have now separated the rapid and gradual changes in 2 sections: 1. Transition in song INIs in the Oceanic Northeast Atlantic region and 2. Gradual changes in song INIs and notes frequencies. Within the first section on rapid song changes (1999 – 2005) we have split the old version Figure 2 into two figures, now Figure 1 only referring to the temporal variation in the SE location; and Figure 2. Referring to the spatial pattern of song INIs in the ONA region (2002/2003). We have also clearly specified these different analyses in the methods, in the section now titled “Temporal and spatial analysis of fin whale song INIs.

6. The points about conformity on L49 could be better supported by citations – in fact, this is not a well-known definition of conformity – the cultural evolution literature is replete with them – look to papers by e.g. Morgan, Whiten, Laland – and the learning of nearby songs over distant ones can be explained simply by the physics of transmission loss under the reasonable assumption that the clearer you hear a song the more likely you are to learn it. It still leads to interesting patterns of variation and does not argue against social learning still underlying song variation, but we are less convinced by this conformity argument.

The manuscript is now not focused on conformity and we have removed most of the references to this process.

7. In places it sounds like the authors are writing from a slightly defensive posture as if they feel bad somehow that fin whale songs are less 'complex' than, say, humpbacks. We encourage the authors to be specific about what is meant by complexity (e.g. number of different song unit types, or number of phrase types in a song), reflect on whether the relationship between complexity and interpretability is as straightforward as they suggest, and finally reflect on why they apparently buy into the idea that 'complexity=interesting' assumption that they feel the need to defend their study against, and revise accordingly.

We have done a major rewrite of the introduction and we do not make this comparison between simple and complex. The motivation of the study has slightly changed, and we do not focus on the advantages of studying simple songs but on the study of song evolution in general (please see response to question 1).

8. L75 "studies on fin whale song changes have been merely descriptive"- how is this study not also descriptive? Reviewers disliked the descriptor 'merely' for descriptive studies – try posing a hypothesis without leaning on some form of description of your study system! This might be better focussed on some strengths of this study like temporal and spatial coverage.

We acknowledge this statement was inaccurate and inappropriate. With the major re-writing of the introduction and modification of motivations, this statement has been removed. As the reviewer suggest, we have focused on the temporal and spatial coverage of our study as a major strength.

9. L79-80 the phrase 'unknown mechanisms of vocal behaviour' is a bit mysterious please be more explicit in your meaning here.

We have done a major rewrite of the introduction and removed the mention of 'unknown mechanisms of vocal behaviour'.

10. L77 this para was vague on the motivations of the study – 'report the dynamics' is a kind of place-holder from which it is hard to predict the actual analyses that were run. Why did you suspect there might be such dynamics? What aspects did you think would be subject to geographic and temporal variation, and why?

The introduction has been rewritten and consequently the motivation of the study modified. We do not further include “report the dynamics” in the introduction.

11. The lack of proper inclusion and interpretation of previous studies documenting seasonal and multi-annual changes in fin whale INI needs to be addressed (e.g., Delarue et al. 2009, Morano et al. 2012). For example, Morano et al. 2012 shows seasonal INI variations and fluctuation between 10s and 16s.

These studies are now included in the introduction in general terms “the song inter-note interval (INI) is the most distinctive parameter between regions(ref) and have been used to differentiate stocks and populations (ref)” and are also discussed in the first paragraph of the discussion, lines 396-399.

12. L95 – Figure 1A shows a map, not changes in frequency?

We thank the reviewers for spotting this mistake. With all the new figures and arrangements, this has been changed and corrected.

13. L95 – Two 'songs' are described here as if the reader should know what they are but they haven't been defined yet. Perhaps explain here what you mean by the '19s-INI' song and the rules you used to differentiate between the songs. Reviewers would ask you to start with a strictly factual description of the results before adding interpretations such as 'we named these two different INI periods as two different song types'. We are not sure that these are different songs – there are definitely different rhythms, but actually the units and their order are exactly the same. So they might be better described as distinct 'tempos' or 'rhythms' rather than completely different songs?

We have now changed the term song type by song INI type referring to different rhythms and leaving the term song type for discussion in the Discussion section (see response to question 2).

14. L97 why is only the SE ONA region presented in such detail – what happens at the other recorders concurrently? And why is only 2002/2003 presented for the entire ONA region?

We have now rearranged the Results section and figures to clarify the different analyses. Please see response to question 5.

15. L99-100 – define a 'hybrid song' and how they were identified – would also be interesting to know how the different INIs were distributed over the sequence – first one then the other, or more mixed?

We have included further information about hybrid songs in the Results section in lines 313-315.

16. The text mentions acquisition duty-cycle with a reference to Table S1, but that table does not list the start date-time and end date-time of the sampling. If one goes to Figure S1 showing a color-coded listing of recording effort, the time resolution is in months, and data in this chart is not linked to sample rate or duty-cycle. A reader must be given all the necessary information by which to assess data quality, validity of analysis results and reliability of interpretation.

We acknowledge that more information should be given about the sampling effort and songs. For this reason, we have made the following changes and additions: (a) added a paragraph in the Results sections about sampling effort and contributions of each region to the total gross data; (b) added a table (Table 1) showing sampled periods by region and location, the duty cycle and sampling rate and total hours of the recordings, number of INIs and HF note frequencies measures as well as their contribution to the total analysed data; (c) Modified the old supplementary Figure S1 (now Figure 5 —figure supplement 1) to show singing season instead of the normal calendar year and added two columns with days analysed and number of INIs and notes; (d) added a new table in supplementary material (Supplementary file 1a) with information on dates, start and end time, duration and number of INIs analysed for each song of each location.

17. Along similar lines: L300-301 – how was the 'lowest SNR' channel selected – from measurements, estimations, sampling? How was SNR measured? Important details such as duty cycle and sampling rate are in a supplementary table, but I think they are important enough to earn their place in the main text.

In the previous version of the manuscript, SNRs were chosen subjectively by looking at spectrograms. We have reanalysed the data, measured SNRs for each note and selected only notes with SNR > 5 dB. We have specified the methodology in the methods section “Song selection criteria” where we first mention the term SNR. We have also added a table in the main text (Table 1) with information on duty cycle and sampling rates for each sampled region and location (See response to question 16).

18. L177 this sentence confused the reviewers – it seems to contradict itself?

We have removed this sentence as it was not adding any new information.

19. L241-242 – isn't the adoption of a new song actually evidence against conformity? Since in a conformist scenario, the new song has to get common enough to be the majority option?

We have changed the focus of the manuscript and the major topic is not conformity, so we have removed most references to this process (See response to question 1).

20. L259-260 – We urge the authors to think more critically about the suggestion that humpback song could function as a sonar (for example, considering the long-standing arguments of Au and Cato against this notion) but it is worth noting that if any baleen whale song were to function in some kind of sonar way then they would be better adapted in terms of their regular structure – however, the frequencies being so low really do make it an unlikely function.

We have reviewed the literature on this topic and acknowledge this is an unlikely function for humpback whale songs. Consequently, we have removed this sentence from the discussion.

21. Text concludes that there was a "complete turnover" of INI. This is a rather strong conclusion given the spatial and temporal sparseness of the data. Is this the right form of words to describe the observations?

We have removed “complete” from the sentence because it is rather strong considering our data.

22. The time-warping used in Figure 2A, Figure 3A, B and C is a bit misleading. Excluding time gaps in available data distorts the time series and could obscure an actual structure (tendency.) We suggest remaking Figure 2A such that the time scale is uniformly linear, not dilated as in a form of dynamic time warping. As presented, it is difficult to interpret. For example, what does the 2001-2002 period include, October 2001 through March 2002? But it could be that all these samples are from Nov 2001 through Jan 2002. Maybe monthly tic marks would help? How about horizontal lines at each of the integer INI values? This might help the reader see slight increases in INI slope over time? Figure S1 is an incomplete synthesis of the data set. It would be valuable to have more and better represented details. We suggest rearranging this figure so that the time months are based on the singing season, not the human calendar. This would have the left-to-right 6six month period go from Oct through March, thereby representing the behavior from the species' behavioral biology perspective rather than a human calendar. This simplifies the representation of when and where each of the possible 20 singing seasons is sampled. It would also be beneficial to clearly show how much each of the sites contributed to the INI and frequency samples.

We completely agree with reviewers regarding the time scale in the x-axis of these figures. We have modified all figures displaying time in their x-axis scales (Figure 1C, Figures 3A, B and C, and Figure 3 —figure supplement 2) and they now have a uniform linear time scale with years in axis ticks and titles and horizontal lines at each INI value. Figure S1, now Figure 5 —figure supplement 1, has also been modified and now time is represented as singing seasons instead of the normal human calendar. Also, days and number of INIs and HF notes are displayed in two columns next to the chronogram. A new table (Table 1) has been added to the main text with information on contribution of each location to the total number of INIs and HF note peak frequencies measured.

23. L309-311 – given that two different methodologies were used (manual and LFDCS), was there any test of the two systems on the same recordings to check they are yield comparable results?

The manual and automatic methods were only used to extract days with fin whale song notes. The automatic method (LFCDS) had been used in some data of the Azores (2008-2012) for a previous study and we reused this data here. Then, a manual inspection of these potential days was done on all datasets to select songs. We have now better explained this in the methods sections.

24. L315 specify how long a 'long series' was.

Long series referred to a minimum of 10 notes, which is now specified in the methods section.

25. The text and figures are not always consistent. For example, the text refers to the E region as the "Bay of Biscay and Iberian Coast", when it contained only a single sensor deployed off the southwest of the Iberian Peninsula. This contrast between the realities of where and when acoustic samples were collected and how the aggregated results are spatially or temporally interpreted or labelled persists throughout the manuscript.

We have moved methods before results to better clarify regions and locations in the first place and be consistent throughout the manuscript when naming locations. We have now adopted our own delimitations of locations and regions and the old ICES Region E “Bay of Biscay and Iberian Coast” is now refereed as SW Iberia. Now this is clearly specified in the methods and the map’s figure caption, where each region has their locations in brackets.

26. Supplementary material needs a table listing the details of the acoustic data recording effort (e.g., from date-time, to date-time) and the acoustic analysis effort (e.g., total recording hours per deployment, for each deployment at each site list the proportion of the total hours relative to the grand total of recording hours, INI-song dates, song time of day start, duration, number of notes.) For example in Figure S1, in January of 2003 for the SE site in the Oceanic Northeast Atlantic region, a song from each of four different days was analyzed yielding a total of 556 INI values for that month. Data from this SE site is the dominant source of recordings in this study. Section 1 in the Supplemental Material is headed in the right direction of addressing sampling aliasing influences on outcome, and this type of effort needs to be included given the large differences in sampling effort/site.

We have added a new table (Table 1) in the main text with information on sampling period, recording hours and contribution of each location to the total number of INIs and HF note peak frequencies measured. We have also added a new table in supplementary material (Supplementary file 1a) with information on date, times of first and last note, duration and number of INIs measured for each fin whale song analysed.

27. We could not find any details on analysis parameters except for the spectrogram figure in Figure 1, where it gives 1024-point FFT, Hann window, 50% overlap. There was no mention of time and frequency resolutions. E.g., if using 1024-point FFT, Hann window, 50% overlap, then the time resolution is 0.26 s and frequency is 1.95 Hz. These resolutions determine the ability to measure and track changes in INI and note peak frequencies. We may have missed seeing a listing of the measurements collected using Raven, but if they weren't there, then they should be included.

The frequency and time resolution are now incorporated in the main text. In the methods section “Measurement of song parameters: inter-note intervals and peak frequencies” we have included in brackets, information on time and frequency resolution: 1.25 Hz frequency resolution and 0.4 sec.

28. L335-337 – the measurement of INIs is insufficiently explained – the figure implies this was a separate measurement from the Raven selection boxes, yet there are robust measurement methods in Raven for measuring onsets based on energy content over time (e.g the 5% cumulative energy threshold).

In the previous manuscript version, we had measured INIs by calculating the time difference between two begin times of consecutive 20-Hz notes. As the reviewer points out, this was not a robust measurement as it highly depends on the selection box. For this reason, we have reanalysed the entire dataset with Raven to calculate INIs using the 5% cumulative energy threshold, as the reviewer suggests. This analysis has not changed our results. We have also included a description of all Raven measurements in a new table (Supplementary file 1c) placed in supplementary material.

29. L350 song types were known prior to this study? Otherwise, this is a result in the wrong place…

We completely agree with the reviewer and have now renamed song types with INI types (please see response to comment 2).

30. L360 – 'except for the Barent's Sea which showed different INIs' – this is a little problematic since it implies you excluded data not for any a-priori reason but because it didn't fit with your preferred model – it is obvious why such a process would not be conducive to rigour. If there is reason to believe the whales from that region belong to an entirely different population and thereby establish an a-priori reason for their exclusion then that should be presented and justified, but excluding data from analyses after results are known isn't a robust way to go about things.

We agree with the reviewer and acknowledge this was not scientifically correct. In consequence, we have removed this sentence and fitted the linear model considering all regions sampled and recalculated the changing rate.

31. The degree to which this is evidence of vocal learning may be a bit overplayed. Definitely there is change, but it is tricky to compare this to e.g. experimental demonstrations. For example, age-related changes in a changing post-whaling demographic scenario should at least be considered?

We lowered the tone when discussing vocal learning in fin whales, but we still believe there is enough evidence from previous studies that this species is able of vocal learning. We do not affirm that our study proves vocal learning in this species, but we discuss it considering all available evidence of this process (other studies). As the reviewer suggested, we have added a discussion on the age-related changes in a changing post-whaling demographic scenario as a potential driver of song changes in fin whales.

32. The document contains various grammatical and punctuation errors, and there are several errors in the reference section. Please proof-read the manuscript thoroughly.

We have proof-read the manuscript thoroughly.